# Breast cancer remodels lymphatics in sentinel lymph nodes

Dominik Eichin [1,2,10], Diana Lehotina [1,2,10], Anni Kauko [2], Maki Uenaka[1], Meri Leppänen [1], Kati Elima[1,2,3], Minna Piipponen [4], Tapio Lönnberg [2,4], Pia Boström[5], Ilkka Koskivuo[6], Tero Aittokallio [2,7,8,9], Maija Hollmén [1,2], Akira Takeda [1,2,11] ✉ & Sirpa Jalkanen [1,2,11] ✉

Cancer metastasis to sentinel lymph nodes (LNs) is often the first marker of potential disease progression. Although it is recognized that tumor-induced lymphangiogenesis facilitates metastasis into LNs in murine models, tumor-induced alterations in human lymphatic vessels remain obscure. Here we use single-cell RNA sequencing and high-resolution spatial transcriptomics to profile lymphatic endothelial cell (LEC) subsets in paired metastatic and non-metastatic LNs obtained from female patients with treatment-naïve breast cancer. Tumor metastasis decreases immunoregulatory LEC subsets, such as PD-L1+ subcapsular sinus LECs, while inducing an increase in capillary-like CD200+ HEY1+ LECs. Matrix Gla protein (MGP) is the most upregulated gene in metastatic LN LECs, and its expression on LECs is TGF-β and VEGF dependent. Upregulated MGP promotes cancer cell adhesion to LN lymphatics. Thus, breast cancer cell metastasis to LNs remodels LEC subsets in human LNs and escalates MGP expression, potentially facilitating cancer cell dissemination through the lymphatic system.

Lymphatics play a critical role in the immune system by efficiently transporting antigens and immune cells into lymph nodes (LNs), which allows the timely initiation of an immune response or tolerance toward antigens. However, this transportation network can be exploited by cancer cells, contributing to their rapid dissemination. Cancer cells metastasize to draining LNs through lymphatics, colonize LNs and induce immune tolerance against tumor antigens[1-3]. LN metastasis also protects cancer cells from oxidative stress in subsequent systemic dissemination[4]. In humans, metastasis of sentinel LNs is a critical parameter for predicting patient mortality[5-7]. Therefore, understanding the mechanisms by which cancer cells migrate to LNs and promote metastatic tolerance is of vital importance.

Cancer cells are highly motile, and they manipulate other cell types to facilitate effective metastasis. One well-known mechanism is tumor-induced lymphangiogenesis. In primary tumors, cancer cells or other cells, such as macrophages, secrete VEGF-C, which induces the proliferation and sprouting of lymphatic endothelial cells (LECs). This allows cancer cells to metastasize into LNs more effectively[8-10]. Another lymphangiogenic growth factor, VEGF-D, is secreted from primary tumors and regulates the dilation of collecting lymphatic vessels and subsequent metastasis by regulating prostaglandin generation[11]. Furthermore, in addition to lymphangiogenesis and the promotion of cancer cell spreading in primary tumors, there is evidence that cancer cells can modulate lymphatic vessels in the draining LNs and thereby facilitate their arrival[12-14]. Thus, the interaction

[1]MediCity Research Laboratory, University of Turku, Turku, Finland. [2]InFLAMES Flagship, University of Turku, Turku, Finland. [3]Institute of Biomedicine, University of Turku, Turku, Finland. [4]Turku Bioscience Centre, University of Turku and Åbo Akademi University, Turku, Finland. [5]Department of Pathology, Turku University Hospital, Turku, Finland. [6]Department of Plastic and General Surgery, Turku University Hospital, Turku, Finland. [7]Oslo Centre for Biostatistics and Epidemiology, OCBE, Faculty of Medicine, University of Oslo, Oslo, Norway. [8]Institute for Cancer Research, Oslo University Hospital, Oslo, Norway. [9]Institute for Molecular Medicine Finland, FIMM, HiLIFE, University of Helsinki, Helsinki, Finland. [10]These authors contributed equally: Dominik Eichin, Diana Lehotina. [11]These authors jointly supervised this work: Akira Takeda, Sirpa Jalkanen. ✉e-mail: akira.takeda@utu.fi; sirjal@utu.fi

between LECs and cancer cells is well recognized, but far from deciphered. In addition, many studies have relied on murine models and overlooked LEC heterogeneity, leaving it unclear how cancer cells alter LEC subsets in humans.

Recent studies employing single-cell RNA sequencing (scRNA-seq) have revealed LEC diversity in various organs such as the LN, skin, and intestine[15–19]. LN LEC subsets are located in distinct areas, such as the subcapsular sinus (SCS) and medullary sinus, and perform subset-specific functions[20]. LN LECs play crucial roles in regulating immune response through multiple mechanisms, such as controlling immune cell migration, transporting and storing antigens, and presenting antigens to immune cells[20,21]. Although LEC heterogeneity has been described by us and others in healthy conditions[15–17] it has not been studied in detail in diseases such as inflammation and cancer. A recent study using scRNA-seq demonstrated that skin inflammation induces transcriptional changes in SCS floor LECs in mouse LNs[16]. However, it remains largely unknown how tumor metastasis to LNs affects LEC subsets in humans and how it impacts cancer metastasis and tumor immunity.

Here, we investigated LEC subsets in paired metastatic and non-metastatic LNs from patients with treatment-naïve breast cancer using scRNA-seq. By analyzing paired samples, we were able to detect tumor-associated changes in human LN LECs. We identified previously uncharacterized LEC subsets accumulating in metastatic LNs and transcriptional changes in established LEC subsets. Matrix Gla protein (MGP) was one of the most upregulated genes in all LN LEC subsets across all the patients. We further confirmed MGP upregulation on LECs in in vitro cocultures with breast cancer cell lines and in the presence of the conditioned medium (CM). We also analyzed the factors behind this upregulation and the consequences of increased MGP on the behavior of LECs.

## Results

### Comparative single-cell analysis of LECs in human LNs

To understand how cancer cell metastasis impacts LECs in metastatic LNs, we performed scRNA-seq of LECs isolated from metastatic LNs as well as non-metastatic, distant LNs that do not have tumor cells. These samples were obtained from patients with breast cancer undergoing mastectomy with axillary node clearance. LECs were enriched by depleting CD45+ cells from single-cell suspensions and subsequently sorting podoplanin (PDPN)+ CD31+ LECs from distant and metastatic LNs of seven patients with luminal and two with Her2-positive breast cancer (Fig. 1a, b and Supplementary Fig. 1a). The absence or presence of tumor cells in distant or metastatic LNs was confirmed by staining single-cell suspensions with an anti-pan-cytokeratin antibody (Fig. 1a). Pan-cytokeratin positive cancer cells were CD45-CD31-PDPN- (Supplementary Fig. 1b). PDPN protein expression on LECs in metastatic LNs was higher than that in distant LNs, indicating that tumor metastasis alters LECs in metastatic LNs. No increase was seen on fibroblastic reticular cells (FRCs) or blood endothelial cells (BECs) of metastatic LNs (Fig. 1c and Supplementary Fig. 1c). The frequency of LECs in LNs varied, but we did not observe proliferation of LECs in the sentinel LNs (Supplementary Fig. 1d), as reported in a mouse study[22]. To integrate scRNA-seq data collected on different days, we employed Seurat version 4, which facilitates the alignment of shared cell populations across diverse datasets and eliminates technical batch effects[23]. Without the alignment and batch correction, cells from different patients were clustered separately (Supplementary Fig. 2a). The integrative analysis identified PROX1+ LEC subsets, JAM2+ BECs, COL1A1+ stromal cells, PTPRC+ leukocytes, including MZB1+ plasmablasts, KRT19+ cancer cells, and MKI67+-proliferating cells (Supplementary Fig. 2b, c).

To comprehensively characterize LEC subsets, we subclustered PROX1+ LECs from these enriched populations and analyzed 99,671 LN LECs, including 38,663 LECs from distant LNs and 61,007 LECs from metastatic LNs (Fig. 1d). Using unsupervised clustering, we identified

14 clusters within the LEC subsets (Fig. 1d). Notably, some LEC subsets, such as cluster 4, were more abundant in metastatic LNs than in distant LNs. To further characterize the LEC subsets in distant and metastatic LNs, we analyzed differentially expressed genes (DEGs) between the LEC subsets (Fig. 1e, f). All LEC clusters expressed the typical LEC markers PROX1 and FLT4 (also known as VEGFR3) (Fig. 1f and Supplementary Fig. 2c). Clusters 0–2 highly expressed atypical chemokine receptor 4 (ACKR4), which is selectively expressed by SCS ceiling LECs. Cluster 3 highly expressed pentraxin 3 (PTX3), PDPN, SPARC, and neuropilin 2 (NRP2), which are markers of paracortical sinuses in human and mouse LNs. Metastasis-induced LEC cluster 4 abundantly expressed biglycan (BGN), the transcription factors HEY1 and SOX4, chemokine CXCL1 and the immunosuppressive molecule CD200 (Fig. 1e, f). Interestingly, this subset selectively expresses acetylglucoaminyltransferase 5 and 7 (B3GNT5 and B3GNT7), indicating a specific glycosylation pattern on this LEC. Clusters 5 and 6 shared specific marker genes, such as THY1 and CD74, but cluster 5 lacked SCS floor LEC markers, including CCL20, TNFRSF9, and CD274 (also known as PD-L1), which were expressed by cluster 6. Given that bridge LECs (also known as trans-sinusoidal LECs) connect the SCS floor with the ceiling and express both SCS ceiling and floor LEC markers, we assigned clusters 5 and 6 as bridge and SCS floor LECs, respectively. Clusters 7, 8, and 9, which were not found in our previous study[15], expressed annexin A1 (ANXA1), interferon-stimulated gene 15 (ISG15), and catenin beta-1 (CTNNB1, also known as β-catenin), respectively. Since these clusters also expressed ACKR4, they may represent subtypes of SCS ceiling LECs. Annexin A (ANXA1) is expressed in vascular endothelial cells (ECs) of primary solid tumors, yet ANXA1+ LECs were found in both distant and metastatic LNs. Clusters 10 and 11 expressed the valve EC marker claudin 11 (CLDN11). Valve LEC1 and LEC2 clusters express neogenin 1 (NEO1) and secretogranin III (SCG3), respectively, and they correspond to LECs on the upstream and downstream sides of valves[15].

Cluster 12 expressed c-type lectins CLEC4G, CLEC4M, and CD209, which are selectively expressed in LN medullary sinus LECs[15]. Cluster 13 selectively expressed lymphotoxin-β (LTB). Given that this subset expresses significantly both paracortical sinus LEC markers, such as PTX3 and NRP2, and medullary sinus LEC markers, including CLEC4M and legumain (LGMN) (Fig. 1e, f), this subset is likely intermediate LECs between medullary and paracortical sinus LECs. This finding aligns with our previous findings, where using LEC trajectory analysis, we found a close relationship between paracortical and medullary sinus LECs[16]. The LEC subsets in distant LNs were also detected in our published dataset from head and neck LNs by visualizing key marker genes of the subsets (Supplementary Fig. 3). Altogether, we demonstrated the highly heterogeneous composition of LECs in human LNs by analyzing a large number of LECs and found significant changes in metastatic LN LECs of patients with breast cancer.

### LN metastasis remodels LEC subsets

Next, we examined changes in LEC subsets within metastatic LNs. The frequency of certain LEC subsets, such as the SCS ceiling and clusters 0, 1, and 2, remained unchanged in metastatic LNs. However, the frequency of other LEC subsets underwent significant changes. Cluster 3 (paracortical sinus LECs) and cluster 4 (CD200+ HEY1+ LECs) increased in metastatic LNs, whereas cluster 6 (SCS floor LECs), and cluster 12 (medullary sinus LECs) decreased (Supplementary Fig. 4a–c). These alterations in the LEC subsets varied between the patients. However, the trend was consistent across the patients (Supplementary Fig. 4d). Changes were observed in patients with both luminal and Her2-enriched breast cancer (Supplementary Fig. 4d, e). To analyze the differential abundance of cell subsets between distant and metastatic LNs, we used miloR, a scalable statistical framework for differential abundance testing on single-cell datasets[24] (Fig. 2a, b). This analysis

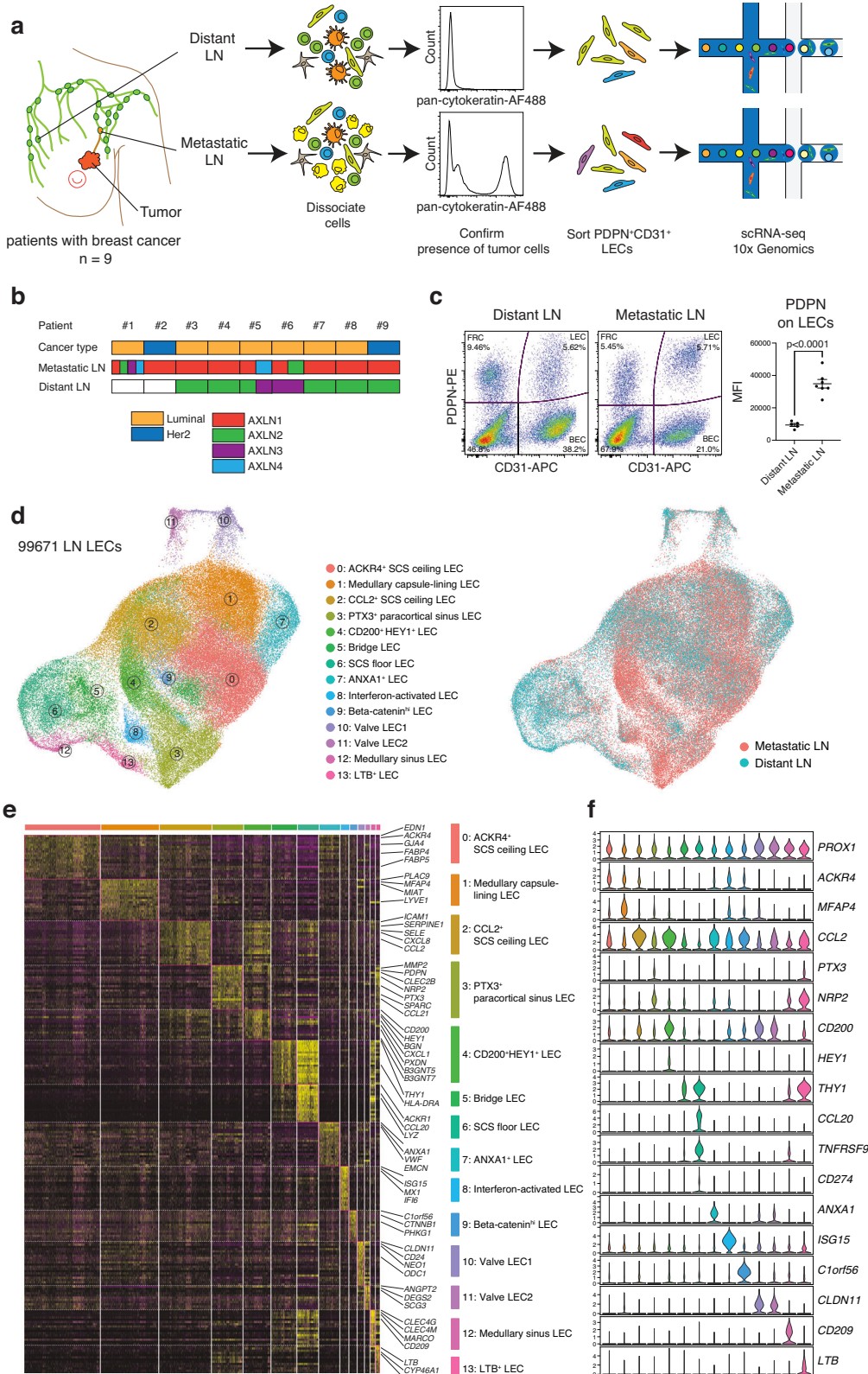

**Fig. 1 | Single-cell analysis of LECs in distant and metastatic LNs from patients with breast cancer. a** Workflow. Distant LNs and paired metastatic LNs from nine individuals were used in this study. Cells were dissociated immediately after surgery, and the presence of tumor cells was verified by staining with pan-cytokeratin. CD45⁺ cells were depleted and PDPN⁺ CD31⁺ LECs were enriched using fluorescence-activated cell sorting. 10x Genomics Chromium scRNA-seq was employed to profile the cells. **b** Sample information. Patients #2 and #9 had Her2-enriched breast cancer and the others had luminal breast cancer. Paired metastatic and distant axillary LNs were collected from patients #1 to #9. **c** PDPN and CD31 staining of distant and metastatic LNs. The mean fluorescent intensity (MFI) represents PDPN expression on LECs (distant LN, $n = 5$ patients; metastatic LN, $n = 7$ patients) (mean ± SEM, two-tailed, unpaired Student's $t$-test). Source data are provided as a Source Data file. **d** UMAP plot of 99,671 LECs from distant and metastatic human LNs, color coded by cluster (left) or metastatic state (right). **e** Heatmap displaying single-cell expression of the top DEGs in LEC subsets, with selected genes labeled. **f** Gene expression differentiating the 14 LEC clusters, illustrated in violin plots. AXLN, axillary lymph node.

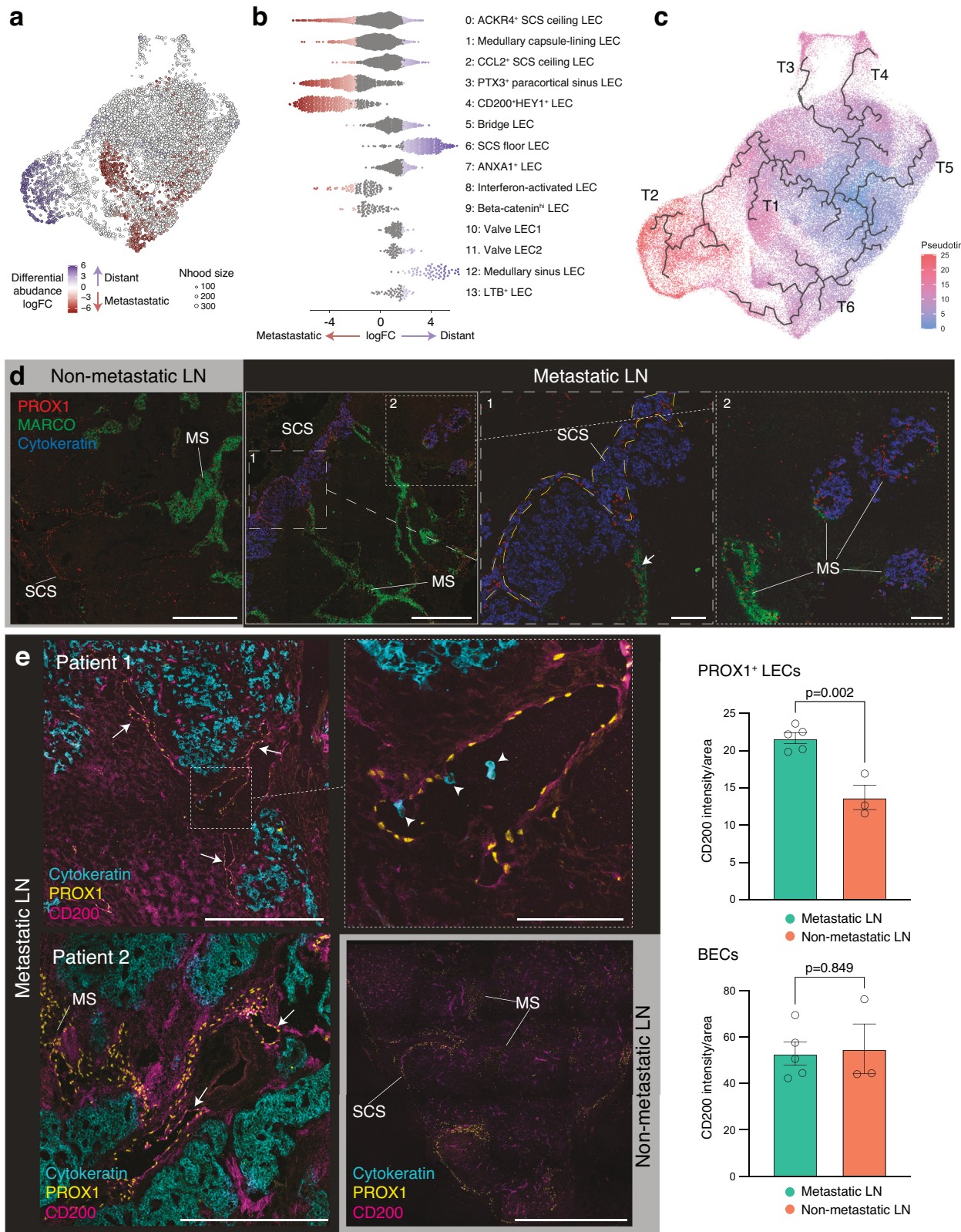

revealed a significant enrichment of cluster 4 (CD200⁺HEY1⁺ LECs) and an increase of cluster 3 (paracortical sinus LECs) in metastatic LNs. In contrast, there was a significant depletion in cluster 6 (SCS floor LECs) and cluster 12 (medullary sinus LECs) in metastatic LNs. The abundance of other subsets did not change significantly (Fig. 2a, b). Moreover, trajectory inference identified six distinct LEC differentiation lineages (Fig. 2c). Among these, trajectory 1 (T1) and trajectory (T2) led to

cluster 4 (CD200⁺HEY1⁺ LECs) and cluster 6 (SCS floor LECs), respectively, both originating from cluster 2 (CCL2⁺ SCS ceiling LECs). This suggests that LN metastasis preferentially drives LEC differentiation towards T1 lineages rather than T2. SCS floor and medullary sinus LECs highly express inflammatory molecules, such as neutrophil chemoattractants, compared with other LECs[15], and maintain sinus macrophages[25], which are important for tumor immunity[26].

**Fig. 2 | Metastasis through lymphatic sinuses alters LEC subsets in human LNs.** **a** Differential abundance testing using miloR. Neighborhoods are colored by their log fold abundance change between distant (blue) and metastatic (red) LNs. Non-differential abundance neighborhoods (FDR >10%) are colored white. **b** Beeswarm plot of the cell subset distribution of log fold change between normal and distant LNs. **c** The trajectories of LN LEC differentiation are shown in a UMAP plot. Six distinct LEC trajectories (T1-T6) were identified using Monocle single-cell trajectory analysis. **d** Immunostaining of cancer cells and LEC markers PROX1 and MARCO in non-metastatic (left) and metastatic LNs (right). Zoomed-in images displaying SCS containing cytokeratin⁺ cancer cells (left) and medullary sinuses (MS), both with and without cancer cells (right). A cancer cell in the MARCO⁺ sinus is indicated by an arrow. Blue, cytokeratin; red, PROX1; green, MARCO. Images are representatives of two individuals with similar results. **e** Immunostaining of CD200⁺ lymphatics and its quantification in metastatic (upper and lower, black background) and non-metastatic (lower, gray background) LNs. CD200⁺ lymphatics and individual cancer cells within lymphatics are indicated by arrows and arrowheads, respectively. Blue, cytokeratin; green, PROX1; red, CD200. Scale bars, 500 μm. Images are representatives of seven individuals with similar results. SCS: subcapsular sinus; MS: medullary sinus; B: B cell zone. Circles in the bar plots represent biological replicates (mean ± SEM, two-tailed, unpaired Student's *t*-test). Metastatic LN, *n* = 5; non-metastatic LN, *n* = 3. Source data are provided as a Source Data file. Scale bars: 500 μm (zoomed-out), 100 μm (zoomed-in).

Immunohistochemical analysis of metastatic LNs showed accumulation of cytokeratin⁺ tumor cells within the SCS, cortex near the SCS, and medullary sinuses (Fig. 2d). We also observed individual tumor cells traveling through MARCO⁺ medullary sinuses (Fig. 2d). Medullary sinuses without tumor cells expressed MARCO, but those containing tumor cells in the same LNs lost MARCO expression, despite maintaining expression of the lymphatic identity marker PROX1 (Fig. 2d), indicating that tumor metastasis through the sinuses alters LEC phenotypes. This finding is in line with the decrease of cluster 12 (medullary sinus LECs) in metastatic LNs (Fig. 2a, b).

We next sought to identify the location of robustly enriched CD200⁺ HEY1⁺ LECs in metastatic LNs (Fig. 2e). CD200 was highly expressed in BECs within both metastatic and non-metastatic LNs at comparable levels. However, CD200 expression on PROX1⁺ LECs was significantly elevated in metastatic LNs compared to nonmetastatic ones (Fig. 2e). Interestingly, PROX1⁺ lymphatics expressing CD200 exhibited a capillary-like lymphatics in metastatic LNs, whereas these lymphatics were undetectable in non-metastatic LNs. Moreover, we observed individual cancer cells within capillary-like CD200⁺ lymphatics, suggesting that cancer cells may disseminate through these lymphatics for systemic spread (Fig. 2e). CD200 is known as an immunosuppressive molecule. To investigate its potential immunosuppressive role in LECs, we co-cultured CD200-expressing human LECs (Supplementary Fig. 5a) with peripheral blood T cells in the presence of a blocking anti-CD200 antibody. CD200 blockade markedly increased the expression of T cell activation markers, including CD69 and CD25, indicating that CD200 on LECs suppresses T cell responses (Supplementary Fig. 5b).

To further characterize remodeling of LECs, we performed high-resolution spatial transcriptomics using Visium HD on non-metastatic and metastatic LNs from two patients (Fig. 3). Tissue sections were stained with anti-PROX1, followed by spatially resolved RNA sequencing at 2 μm × 2 μm resolution. Spots with PROX1 mRNA (aggregated in 8 μm × 8 μm bin) also showed PROX1 protein expression, validating the robustness of the method (Supplementary Fig. 6a). Unbiased clustering using UMAP identified several cell types, including *KRT19*⁺ cancer cells, *CD3E*⁺ T cells, *MS4A1*⁺ B cells, *MZB1*⁺ plasmablasts, *FLT3*⁺ dendritic cells (DCs), and *PROX1*⁺ LECs, which were then visualized within the tissue context (Fig. 3a). Metastatic LNs exhibited extensive cancer cell infiltration and marked architectural alterations, along with DC-T cell association near T cell clusters, suggestive of local anti-tumor immune responses (Fig. 3a). Next, we isolated *PROX1*⁺ LECs across all four samples, integrated them, and performed sub-clustering to resolve LEC heterogeneity (Fig. 3b).

The resulting LEC subsets were then spatially mapped within the tissue (Fig. 3c). In non-metastatic LNs, we identified distinct subsets including *TNFRSF9*⁺ *PD-L1*⁺ SCS floor LECs, *NTS*⁺ SCS ceiling LECs, and *CLEC4M*⁺ medullary sinus LECs (Fig. 3b, d), located at the SCS floor, beneath the capsule and in the medulla, respectively (Fig. 3c). These findings are consistent with our previous work[15] and largely align with the scRNA-seq results (Fig. 3b). In contrast, metastatic LNs showed marked remodeling of the LEC compartment, with notable reductions in SCS floor and medullary sinus LECs (Fig. 3b, c), consistent with

scRNA-seq data. The LN from patient #5 lacked SCS floor LECs but retained medullary sinus LECs to some extent, whereas patient #3's LN lack both subsets almost entirely, suggesting a more advanced stage of metastasis in patient #3. Four LEC types were predominantly detected in metastatic LNs. Two of these (cancer cells and LEC1, cancer cells and LEC2) exhibited high expression of cancer cell-associated genes such as *KRT19*, *CDH1*, *MUC1*, and *TFF1*, suggesting their close interaction with cancer cells. The other two clusters, enriched in metastatic LNs, displayed characteristics of paracortical sinuses, expressing markers such as *PTX3* and *SPARC* (Fig. 3c and Supplementary Fig. 6b). They also expressed *FN1* and *BGN*, which were detected in *CD200*⁺ *HEY1*⁺ LECs. Although the Visium HD data clearly recapitulate key findings from scRNA-seq as above, not all LEC subsets and subset-specific gene signatures identified by scRNA-seq were detectable in the Visium HD, possibly due to the lower transcript capture efficiency. Overall, these data show that metastasis remodels the normal architecture of LN, leading to the emergence of distinct types of LECs, including *CD200*⁺ *HEY1*⁺ LECs and paracortical sinus LECs, while depleting LECs that are important for an immunological response.

## Metastasis-induced transcriptional changes in LN LECs

To investigate transcriptional changes upon tumor metastasis, we identified marker gene expression in neighborhoods corresponding to enriched or depleted LECs in metastatic LNs (Fig. 4a, left). This analysis revealed that enriched cells in metastatic LNs express *MGP*, growth arrest specific 6 (*GAS6*), *BGN*, *PDPN*, transcription factor 4 (*TCF4*), and *CD200*. In contrast, depleted cells express inflammatory genes such as *NFKB-IA*, *CXCL1-CXCL3*, *CD74*, and *CD44* (Fig. 4a, right). Pseudobulk analysis of the scRNA-seq data, irrespective of clusters, also showed the upregulation of *MGP*, *CCL21*, *BGN*, *PDPN*, and microfibril-associated protein 2 (*MFAP2*) in metastatic LNs (Fig. 4b, c). These genes were upregulated in most LEC subsets in metastatic LNs (Fig. 4c and Supplementary Data 1). Immunostaining of MGP revealed its expression on the LN capsule and trabecula but not on LECs in distant LNs and confirmed its upregulation on PROX1⁺ LECs in metastatic LNs (Fig. 4d). MGP expression was also detected in stromal cells and BECs in our scRNA-seq data (Supplementary Fig. 7a). The upregulation of MGP in metastatic LN LECs was further confirmed by our spatial transcriptomics dataset (Fig. 4e).

Gene enrichment analysis of upregulated genes (log₂FC >0.5, *p* < 0.05) in metastatic LN LECs revealed a notable enrichment of genes associated with Gene Ontology terms "collagen-containing extracellular matrix remodeling" and "external encapsulating structure", including *MGP*, *BGN*, microfibrillar-associated proteins *MFAP2* and *MFAP4*, and fibronectin 1 (*FN1*) (Fig. 4f, left). The most downregulated genes in metastatic LNs included superoxide dismutase 2 (*SOD2*) and the inflammatory chemokines and cytokines *CXCL3*, *IL6*, *CXCL2*, and *CXCL1* (Fig. 4a, b). Downregulation of *SOD2* and *CXCL3* was detected in many clusters (Fig. 4c). Gene enrichment analysis of downregulated genes (log₂FC <−0.5, *p* < 0.05) revealed significant enrichment in leukocyte trafficking-associated Gene Ontology terms, including 'leukocyte cell-cell adhesion' and 'leukocyte migration' (Fig. 4f, right). This indicates that proper leukocyte trafficking is impaired in metastatic

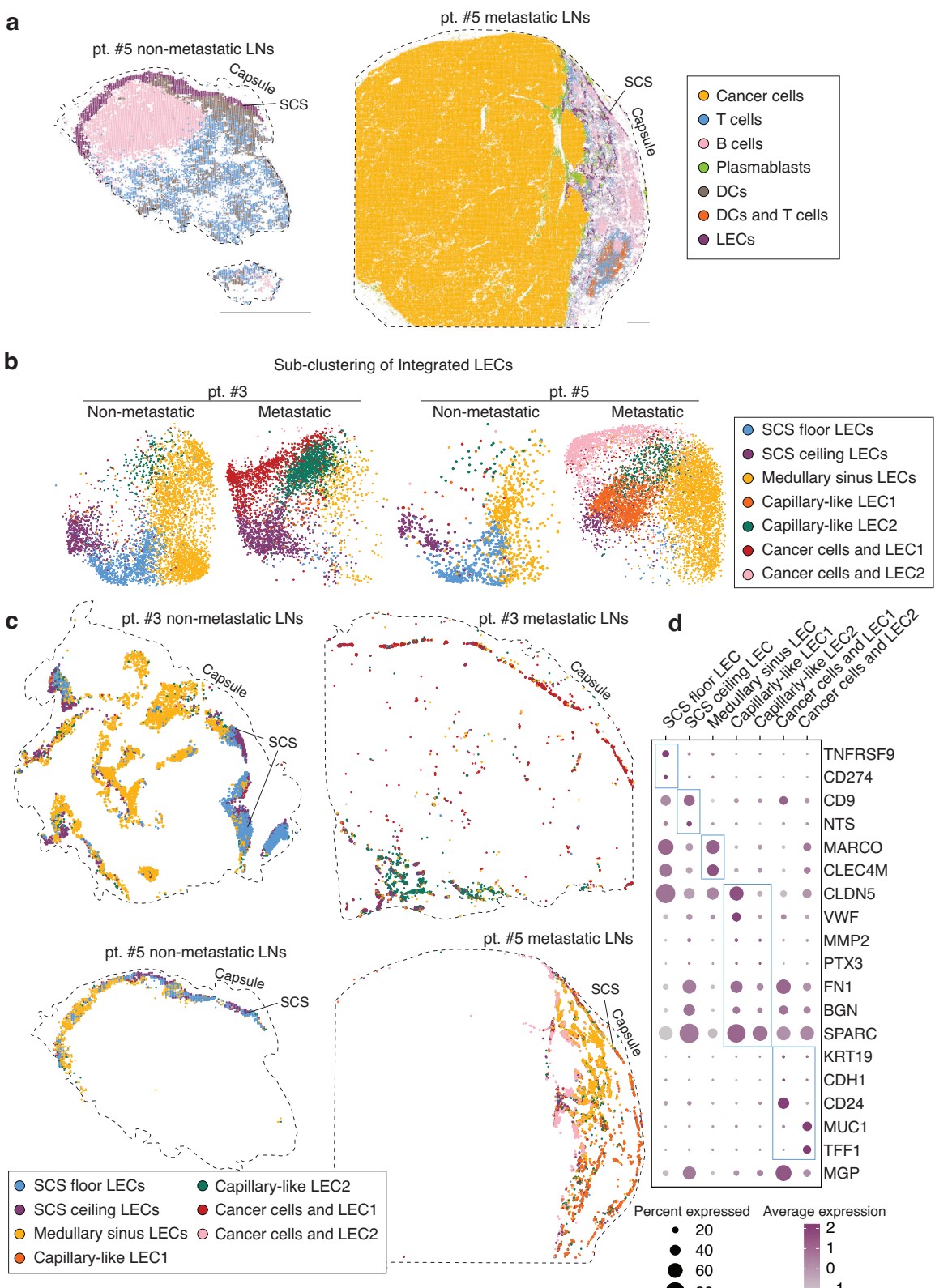

**Fig. 3 | Visualization of remodeled LEC subsets using high-resolution spatial transcriptomics. a** Cellular subsets in non-metastatic and metastatic LNs from two patients (patient #3, 5). UMAP plots were generated from the Visium HD dataset and key cellular subsets, including LECs, cancer cells and immune cell subsets, were identified and spatially visualized within the tissue sections. The capsules were outlined by dotted lines. Scale bars, 500 μm. **b** UMAP showing subclusters of LEC from nonmetastatic and metastatic LNs. PROX1[+] LECs from all samples were selected, integrated, and subclustered. **c** Spatial distribution of the LEC subsets identified in UMAP in both nonmetastatic and metastatic LNs. **d** Gene signatures of LEC subsets found in spatial transcriptomics.

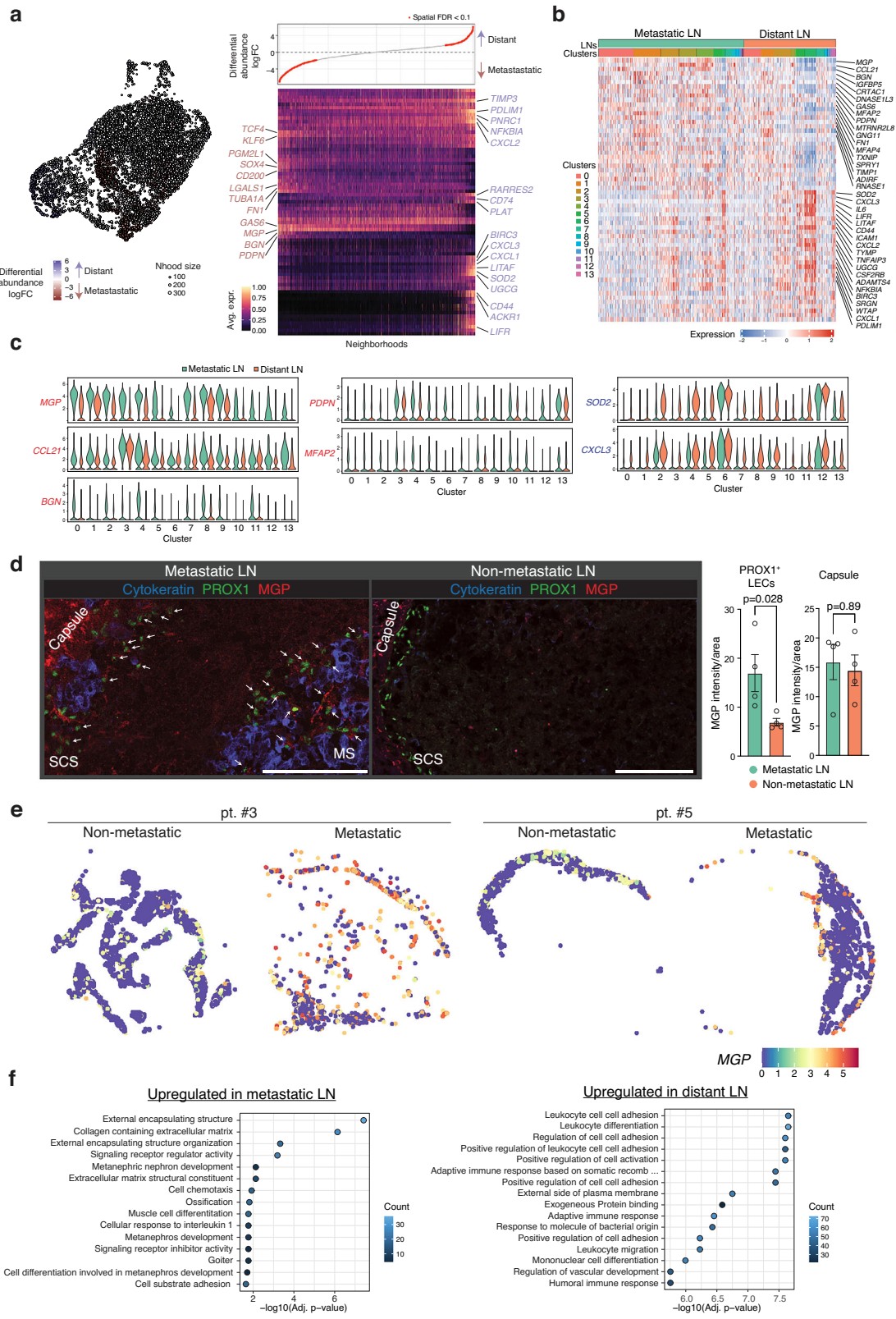

LNs. Notably, *CD274*, which is expressed by SCS floor LECs, was among the genes downregulated in metastatic LNs. *CD274* expression was not detectable in LN cells, including macrophages and BECs, except for SCS floor LECs (cluster 6) (Supplementary Fig. 7a, b).

To investigate whether inflammation induces similar changes in LECs as cancer, we analyzed publicly available scRNA-seq datasets of murine LECs from inflamed LNs[16,27]. LEC subset composition remained largely unchanged in inflamed LNs, and genes upregulated in human metastatic LNs, such as MGP and BGN, were not upregulated (Supplementary Fig. 8a, b). We also examined a recently published scRNA-seq dataset of human LECs from inflamed LNs[28] and similarly found that inflammation did not alter LEC subset composition (Supplementary Fig. 8c). Furthermore, to assess whether these transcriptional changes are also present in primary tumors, we analyzed a publicly

**Fig. 4 | Transcriptional reprogramming of metastatic LN LECs. a** Differential abundance testing using MiloR and a heatmap of differentially expressed genes between differential abundance neighborhoods in LN LECs. The UMAP plot was shown in Fig. 2c and is inserted here for clarity. In the heatmap, columns and rows represent neighborhoods and differentially expressed genes, respectively. Expression values for each gene are scaled between 0 and 1. The upper panel of the heatmap shows the neighborhood log fold change. FDR, false discovery rate. **b** Heatmap showing the expression of the top DEGs between metastatic and distant LNs for each cell. Bars above the heatmap indicate the tissue and cluster origin of each cell (LNs, clusters). **c** Violin plots displaying the top DEG expression between metastatic and distant LNs by cluster, with log-normalized expression value labeled. Nine patients'

samples were integrated for this analysis (**a**–**c**). **d** Immunostaining of MGP and its quantification in metastatic and distant LNs. Zoomed-in images show SCS and medullary sinuses containing cancer cells and MGP expression on PROX1⁺ LECs (arrows). Blue, cytokeratin; red, MGP; green, PROX1. Scale bars, 125 μm. Images are representative of four individuals with similar results. Circles in the bar plots represent biological replicates (mean ± SEM, two-tailed, unpaired Student's *t*-test). Metastatic LN, *n* = 4 patients; non-metastatic LN, *n* = 4 patients. Source data are provided as a Source Data file. **e** MGP expression in LECs of non-metastatic and metastatic LNs detected using Visium HD. **f** Gene Ontology (GO) enrichment analysis of the top DEGs (Wilcoxon rank-sum test, metastatic vs. distant LNs) using the EnrichR package. FDR-corrected *p* values are shown.

available scRNA-seq dataset of ECs from human breast cancer[29] (Supplementary Fig. 9a). We found that MGP expression was higher in LECs within primary tumors compared to those in peritumoral regions both at mRNA (Supplementary Fig. 9b) and protein level (Supplementary Fig. 9c), supporting its relevance in LEC remodeling beyond the LN environment.

### NicheNet intercellular communication analysis predicts the mechanisms of lymphatic remodeling in metastatic LNs

To understand the mechanisms by which LN lymphatics are remodeled upon LN metastasis, we performed NicheNet analyses. This method predicts the link between ligands from sender cells and changes in gene expression in the receiver cells using prior knowledge on signaling and gene regulator networks[30]. In addition to LECs, metastatic and distant LNs contained lymphocytes, macrophages, plasmablasts, cancer cells, non-endothelial stromal and BECs (Fig. 5a). We applied NicheNet to predict which ligand–receptor interactions could drive the DEGs found in CD200⁺ HEY1⁺ LECs and SCS floor LECs, both of which were mostly affected in metastatic LNs. Thus, we designated CD200⁺ HEY1⁺ LECs (cluster 4) or SCS floor LECs (cluster 6) as receivers and the other cell types in LNs as senders that express specific ligands to alter gene expression in the receivers (Fig. 5b). *MGP, PDPN, FN1, HEY1, and SOX4*, which are highly detected in enriched LECs in metastatic LNs (Fig. 4a), were among the predicted target genes in CD200⁺ HEY1⁺ LECs. The top predicted ligands that induce signatures of CD200⁺ HEY1⁺ LECs included TGF-β, VEGF-A, and VEGF-C (Fig. 5c, d).

TGF-β was commonly expressed by multiple cell types (Fig. 5d), including cancer cells, stromal cells and BECs, and its expression was upregulated in metastatic LNs (Fig. 5e), potentially driving the expression of *MGP, FN1, SOX4, PDPN,* and *HEY1* in LECs (Fig. 5c). A transcription factor SOX4 is induced by TGF-β[31,32]. VEGF-A is highly expressed in macrophages and in cancer cells (Fig. 5e), and may induce *MGP* and atypical chemokine receptor *ACKR3*, which is a receptor of CXCL12 (Fig. 5c). CD200⁺ HEY1⁺ LECs express TGF-β receptors TGFBR2 and TGFBR3 (Supplementary Fig. 10), and integrin alpha v ITGAV and integrin beta-1 ITGB1, which can activate a latent form of TGF-β, as well as receptors for VEGF-A and VEGF-C. Notably, TGF-β-dependent LRRC15⁺ cancer-associated fibroblasts[33] were only found in metastatic LNs, indicating that TGF-β signaling is upregulated in metastatic, but not in distant LNs (Supplementary Fig. 11). In contrast, the top predicted ligands that induce signatures of SCS floor LECs in distant LNs included insulin-like growth factor 1 IGF1, Epstein-Barr virus induced 3 EBI3 and LTB, which were mainly expressed by LN stromal cells, macrophages, and lymphocytes, respectively (Supplementary Fig. 12).

### Breast cancer cell-conditioned media alter human LEC transcriptomes

To determine whether tumor-induced changes in LECs could be mimicked in an in vitro system that allows us to further study the observed changes, we used primary human lymphatic endothelial cells (HLECs), which are isolated from human LNs. In particular, our interest was in MGP, which was upregulated in metastatic LN lymphatics of all

patients and whose function on lymphatics has remained unexplored. The HLECs were exposed to conditioned culture media (CM) from four different breast cancer cell lines, MCF-7, T47D, HCC1954, and MDA-MB-231 (Fig. 6a). Both MCF-7 and T47D are estrogen and progesterone positive luminal subtypes of breast cancer, HCC1954 is a HER2-positive one, and MDA-MB-231 represents triple-negative breast cancer cells—the most aggressive form of breast cancer. RNA-seq of tumor-conditioned HLECs revealed changes in the expression of multiple genes, with a similar number of genes being upregulated (101 genes) and downregulated (99 genes). While some genes were uniquely changed following exposure to CM from only one breast cancer cell line, other genes showed the same changes with several or all CM, as shown in the Venn diagrams in Fig. 6b. Although the total number of upregulated and downregulated genes were comparable, MDA-MB-231 cells induced the most unique alterations in gene expression. In addition, MDA-MB-231 cells upregulated more genes than they downregulated, whereas this was the opposite with the CM of T47D cells.

To further evaluate these differential gene expression profiles, we implemented unsupervised clustering of genes that were significantly altered in at least two of the three experimental settings based on the RNA-seq transcriptome data (Fig. 6c). This resulted in clustering according to the CM used, while also revealing a similar induced profile caused by MCF-7 and MDA-MB-231 CM. Volcano plots for each CM are shown in Fig. 6d, where the most altered genes are indicated. They include *MGP, SOCS3, H19,* and *CEBPD* (upregulated) and *ANO9, CX3CL1,* and *VCAM1* (downregulated). In addition, genes associated with a general pro-inflammatory phenotype (shown in red) were downregulated in HLECs cultured with cancer cell CM, whereas anti-inflammatory genes (shown in blue) were upregulated, indicating that soluble factors derived from cancer cells may induce a transition in LECs from a pro-inflammatory to an anti-inflammatory state. To better understand the observed changes in gene expression, we performed a pathway analysis (Fig. 6e). Most significantly, pathways involved with inflammation, such as "cytokine signaling in immune system", "inflammatory response", or "innate immune response" were downregulated. In contrast, the most upregulated pathways include pathways involved in cell adherence and stability, the matrisome, or cellular response to different stimuli (e.g., "PID AJDISS 2PATHWAY (Post-translational regulation of adherence junction stability and disassembly)" or "cellular response to cytokine stimulus").

To verify the observed changes in RNA-seq, we implemented qPCR and flow cytometry assays for selected hits (Fig. 6f, g). This confirmed the overall pattern of expression changes, and among them, MGP, receptor activity-modifying protein 3 (RAMP3), and interleukin 4 inducible 1 (IL4I1) showed the most consistent alterations. The changes in MGP were the most significant with CM from T47D cells and the least with the CM from MDA-MB-231 cells. Moreover, CM of cancer tissue explants resulted in different gene expression profiles than the CM of adjacent normal breast tissue, including a trend to upregulate MGP (Supplementary Fig. 13), suggesting that soluble factors in the tumor microenvironment influence LEC phenotype. We further assessed, using T47D cells, whether direct cocultures cause similar or stronger phenotypic changes than CM. Indeed, direct coculture increased MGP levels significantly more than

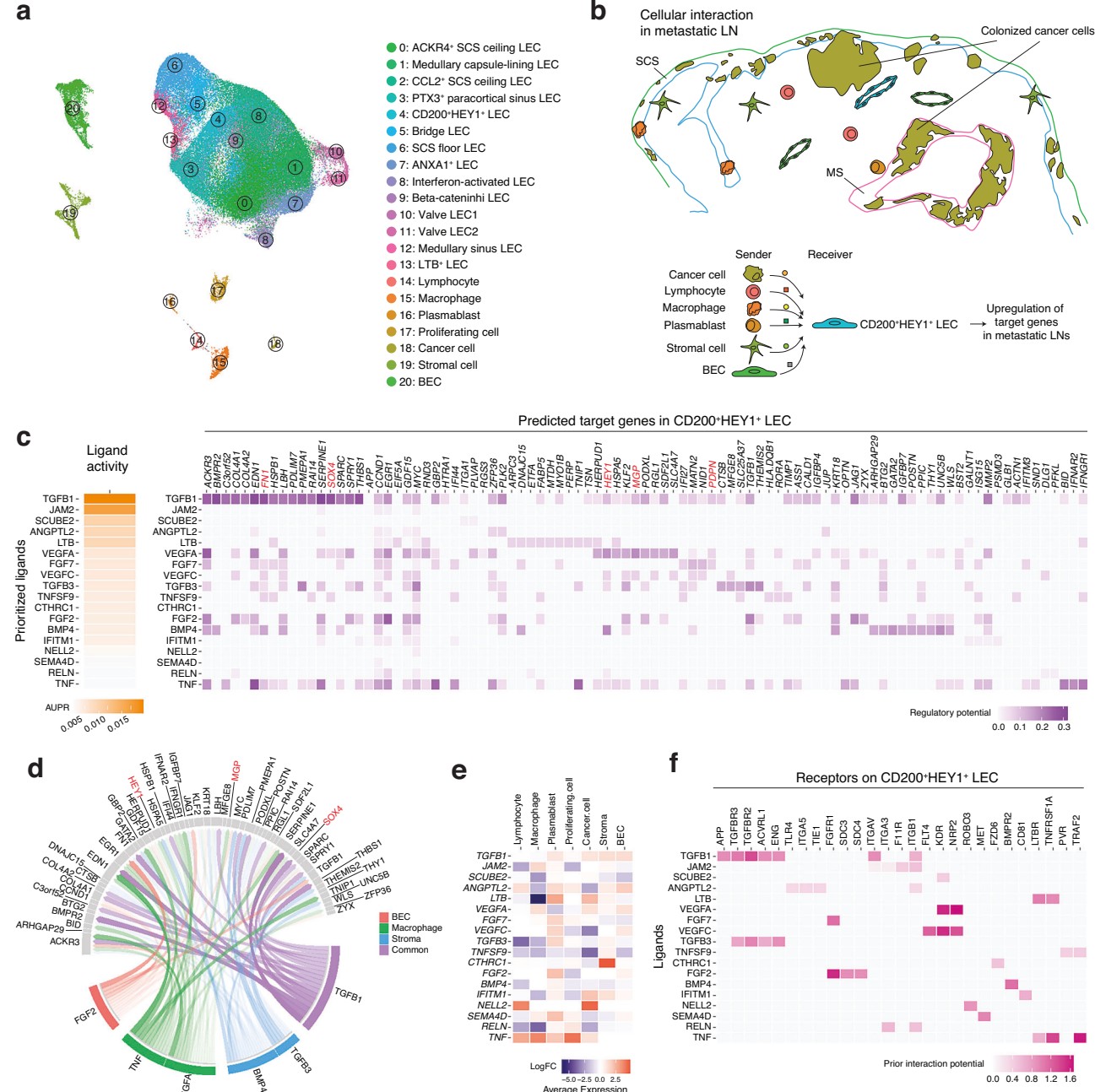

**Fig. 5 | NicheNet predicts the mechanisms driving lymphatic remodeling.**
**a** UMAP of 21 cell subsets, including LEC subsets and other cell types in LNs. **b** A schematic representation of the NicheNet analysis of intercellular communications that induce target genes in CD200⁺ HEY1⁺ LECs in metastatic LNs. CD200⁺ HEY1⁺ LECs were set as the receiver and the other cell types in LNs were set as the sender. **c** Predicted top ligands and their target genes in the CD200⁺ HEY1⁺ LECs. AUPR: area under the precision-recall curve. **d** Circos plot showing the links between predicted ligands from LN cells and their potential target genes in CD200⁺ HEY1⁺ LECs. Low-weight links were removed for clarity. **e** Relative expression of predicted targets in metastatic or distant LNs. **f** Potential receptors expressed by CD200⁺ HEY1⁺ LECs associated with each predicted ligand. Nine patients' samples were integrated for this analysis (**a**–**f**).

CM, whereas comparable changes were not observed for PDPN (Fig. 6h). MGP was, as in our single-cell-sequencing data, again one of the molecules with the most significant changes. Although certain tumors can induce LEC proliferation in murine models, the CM of the cell lines tested did not induce LEC proliferation in vitro (Supplementary Fig. 14).

Blood vessel endothelial cells transform into mesenchymal cells in certain pathologies such as cardiac fibrosis, atherosclerosis, vascular calcification, and cancer. This process is known as the endothelial-to-mesenchymal transition (EndMT) and may play a role in angiogenesis, generation of cancer-associated fibroblasts, and cancer metastasis[34]. We sought to determine whether this process

was induced in LECs by exposure to the CM. As seen in Fig. 6i, this was cell-line specific as EndMT indicative genes *CDH2*, *CDH5*, and *SNAI1* were not altered in HLECs following exposure to CM from MDA-MB-231, MCF-7, and T47D, while *CDH5* and *SNA I1* were upregulated by HCC1954 CM.

**Tumor-induced transcriptional changes in LECs are mediated in part by VEGFs and TGF-β**
Next, we searched for cancer cell-derived factors that can induce transcriptional changes in LECs. The prime candidate cytokines for testing were selected from the published list of cytokines secreted by

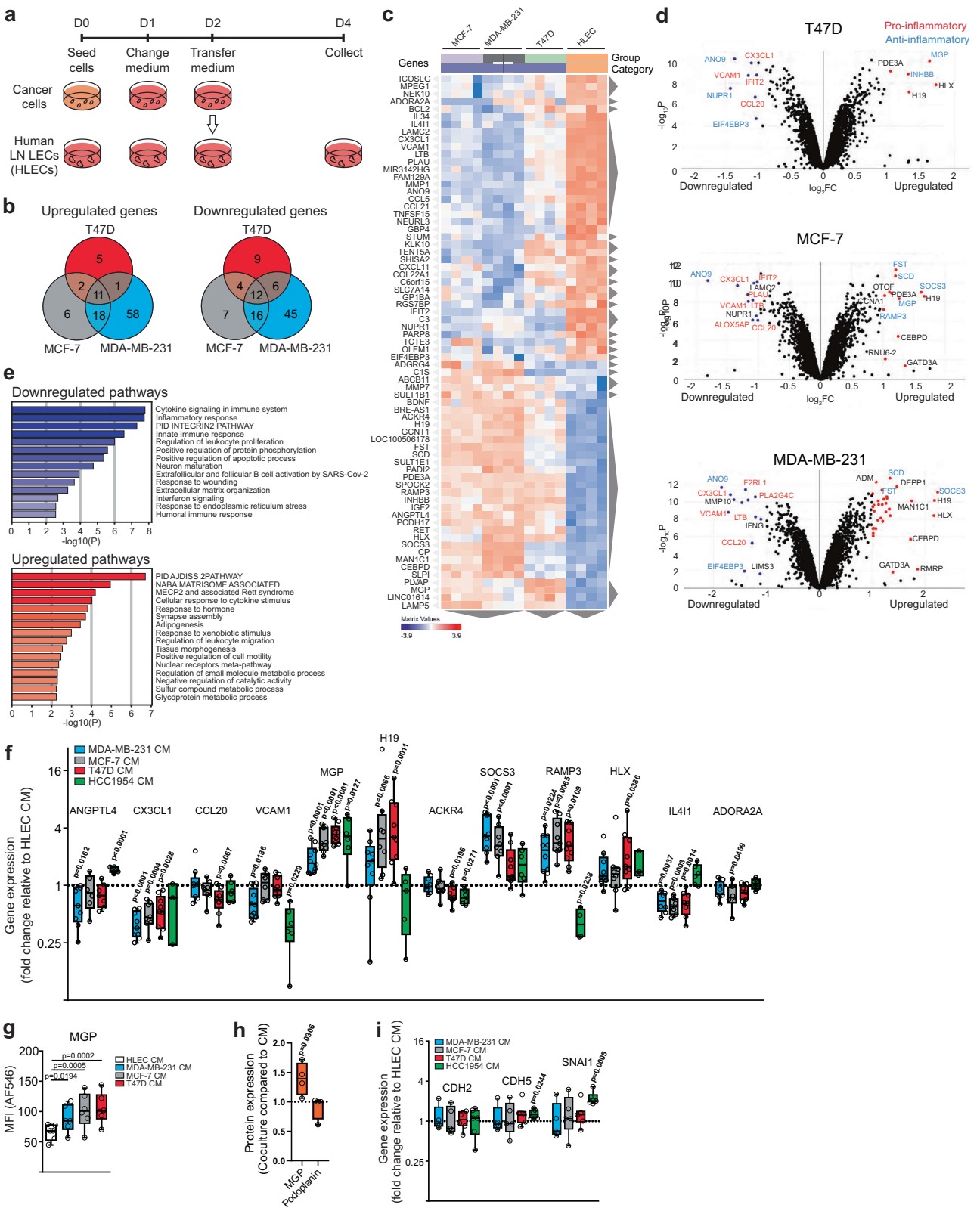

the breast cancer cell lines used in our study[35,36]. The selected candidate cytokines were those secreted by all the lines. To verify their effect, we tested whether recombinant VEGF165 (a major spliced isoform of VEGF-A), VEGF-C, TGF-β, EGF, GM-CSF, M-CSF, or G-CSF affects MGP expression. TGF-β, VEGF-A, and VEGF-C are among NicheNet-predicted ligands that may induce lymphatic remodeling. Indeed, VEGF165, VEGF-C, and TGF-β dose-dependently increased MGP (Fig.

7a). The other cytokines did not have an effect (Fig. 7a and Supplementary Fig. 15). While certain of these changes could be observed with the genes RAMP3 and IL4I1, they also have their specific regulators different from MGP (Supplementary Figs. 16, 17).

We also added blocking antibodies against VEGFR3, TGF-β, EGF, PDGF, and adrenomedullin to cancer cells before starting CM generation, which was subsequently transferred to the HLECs. The addition of

**Fig. 6 | RNA-seq of human LECs exposed to breast cancer cell CM. a** Experimental setup for the in vitro experiments. Cells were seeded at Day 0 (D0), CM was generated on D1 to D2 (24 hours) and receiving cells were exposed on D2 to D4 (48 h). **b** Venn diagrams showing the number of upregulated and downregulated genes obtained by RNA-seq of CM-exposed LECs. **c** Heatmap showing the top DEGs in the CM-exposed LECs. Gray triangles indicate the result of unsupervised clustering for genes and cell types behaving in a similar way. **d** Volcano plots depicting genes upregulated and downregulated in LECs after CM exposure (two-sided Wald test). Genes indicated in red are considered mostly pro-inflammatory, whereas genes in blue are anti-inflammatory. **e** Pathway analysis showing the most significantly downregulated and upregulated pathways (the integrated two-sided Fisher's exact test). LECs are from 4 different sources ($n = 4$) (**a**–**e**). **f** Gene expression changes in

CM-exposed LECs ($n = 3$–9 HLECs). Data were analyzed with a one-way ANOVA. **g** Protein expression levels of MGP in LECs following CM exposure ($n = 6$ HLECs). Data were analyzed with a one-way ANOVA linear mixed model and Dunnett's correction. **h** Protein expression of MGP and podoplanin on LECs after coculture of T47D cells with LECs compared to culture with CM. Data were analyzed with two-way ANOVA and Sidak's multiple comparison test ($n = 4$ HLECs). **i** mRNA Expression of *CDH2*, *CDH5*, and *SNAI1* in CM-exposed LECs. Data were shown as box plots ($n = 5$–7 HLECs). Data were analyzed using a two-way ANOVA linear mixed model and statistical significance was adjusted with Dunnett multiplicity correction. Source data, non-significant $p$ values and detailed experiment and $n$-numbers (biologically independent samples of cultured cells) are provided in the Source Data file.

blocking antibodies against VEGFR3 and TGF-β, led to reduced induction (or reduction) of MGP in the presence of MDA and T47D CM (Fig. 7b). In contrast, we observed no statistically significant changes when blocking antibodies targeting EGF or PDGF were used (Fig. 7b and Supplementary Fig. 15). Similarly, the PLA2 inhibitor AACOCF3 did not alter MGP expression. Instead, the STAT1 inhibitor fludarabine dose-dependently decreased MGP in T47D CM, and inhibition of adrenomedullin increased MGP in MDA CM (Supplementary Fig. 15). We further confirmed that cancer cell-secreted VEGFs are responsible for the effect by using culture medium without VEGFs (Fig. 7c). These observations are in line with NicheNet analysis results of scRNA-seq data (Fig. 5) and indicate that the cytokine milieu in metastatic LNs could play a crucial role in shaping the transcriptional programs of LECs.

We then focused on MGP, because it was upregulated in the sentinel node LECs of all patients and in the HLECs by CM of all breast cancer cell lines in vitro. Moreover, its expression has been associated with the outcome of several cancers[37], but its functional role in lymphatics has remained unexplored. We tested whether MGP could regulate its own expression levels. Therefore, we used an anti-MGP antibody while generating CM from T47D cells, before exposing HLECs to it. As seen in Fig. 8a, this did not have a noticeable effect because MGP expression levels were still elevated. Similarly, also exposing HLECs directly to an anti-MGP antibody or a recombinant form of the protein did not alter its mRNA expression (Fig. 8a, b).

### MGP is involved in cancer cell adhesion to LECs

Because MGP expression in HLECs was upregulated by cancer conditioning in both the patients' scRNA-seq data as well as in our in vitro assays, we wanted to determine the functions of MGP on HLECs. We used siRNA to silence it and could reduce its expression at the RNA level by ~85% (Fig. 8c). It has been shown that MGP inhibits EndMT in the blood vasculature by inhibiting bone morphogenetic protein[38]. Thus, we next tested whether MGP silencing induced EndMT. This was not the case as the observable small changes in gene expression were contraindicative (Supplementary Fig. 18). Using the silenced cells, we performed tube-forming assays, which revealed that the tube-forming properties of control and MGP-silenced HLECs were identical (Fig. 8d). In contrast, when we performed scratch assays, faster migration of MGP-silenced HLECs was observed (Fig. 8e), indicating that MGP inhibits LEC migration.

The impact of MGP on LEC functions was also observed in our cancer cell adhesion experiments. To assess whether MGP can mediate cancer cell adhesion, we examined the binding of recombinant MGP to T47D cancer cells in vitro and found that soluble MGP bound significantly to these cells (Fig. 8f). Furthermore, we tested the role of MGP on cancer cell adhesion to LN lymphatics using metastatic and non-metastatic lymph node sections of four patients and T47D cells in ex vivo adhesion assays. In these assays, the anti-MGP antibody significantly inhibited T47D cell binding to lymphatic sinuses in metastatic LNs but not those in non-metastatic LNs (Fig. 8g).

Overall, these data indicate that breast tumor-induced MGP upregulation on LECs may play roles in inhibiting LEC migration but

increases their interaction with breast cancer cells. Thus, TGF-β and VEGF-induced upregulation of MGP in LECs may be a part of the dissemination mechanism of breast cancer cells in patients.

## Discussion

In this study, we observed significant alterations in LN lymphatics draining breast cancer using scRNA-seq analyses. Metastatic LNs showed a loss of specific LEC subsets, such as the SCS floor and medullary sinus LECs. However, they displayed the emergence of certain abnormal LEC subsets, including CD200+ HEY1+ LECs. This LEC remodeling signature was consistently observed in multiple patients. Among the upregulated molecules, MGP expression was notably high in metastatic LN LECs of patients with breast cancer, a characteristic that was also observed in cultured LN LECs in the presence of CM of breast cancer cells. MGP silencing and inhibition in LECs increased LEC migration and reduced cancer cell adhesion, respectively. These findings demonstrate that lymphatic remodeling in sentinel LNs is intimately linked to cancer progression.

In previous studies, we reported the heterogeneity of LECs in human LNs, identifying six distinct types of LN LECs[15]. In the present study, we discovered additional subsets (a total of 14 clusters) in non-metastatic LNs. This disparity may stem from several factors. Notably, our current report incorporates the analysis of a larger number of LECs, uses a newer, improved version of 10x Genomics kits and Cell Ranger version 3 instead of v2. We also used the updated genome reference GRCh38, in contrast to GRCh37 used in the previous study. This has helped to align the sequenced reads more accurately. For example, PECAM1 was not detected in the previous dataset, but was successfully detected in the current dataset. We currently also employ a higher number of principal component analysis (PCA) dimensions to identify transcriptionally distinct subsets compared with our previous analysis. Moreover, we utilized the more recently developed Seurat version 4 anchor-based integration tools, which surpass the capabilities of the canonical correlation analysis (CCA)-based data integration tools used in our prior study (Seurat version 2.3)[15]. These tools allowed us to apply a higher number of PCA dimensions while effectively mitigating technical batch effects between the samples. Additionally, since the clustering parameters were selected manually to identify 14 LEC clusters, there may be additional cellular subsets or states that have not been captured.

In metastatic lymph nodes, there was an increase in capillary-like CD200+ HEY1+ LECs and paracortical sinus, whereas SCS floor and medullary sinus LECs were reduced. Trajectory analysis suggested that CD200+HEY1+ LECs and SCS floor LECs are transitioned from CCL2+ SCS ceiling LECs, and LN metastasis drive the transition to CD200+HEY1+ LECs. As we described in the previous study[15], we considered the ACKR4+ SCS ceiling as the root of the trajectory by considering that the SCS is generated during LN development following the engulfment of LN anlage by collecting lymphatic vessels[39], which are ACKR4+. Based on this, ACKR4+ SCS ceiling LECs arise early in the formation of the LN lymphatic network, differentiate into multiple LECs depending on the local microenvironment, and transition to

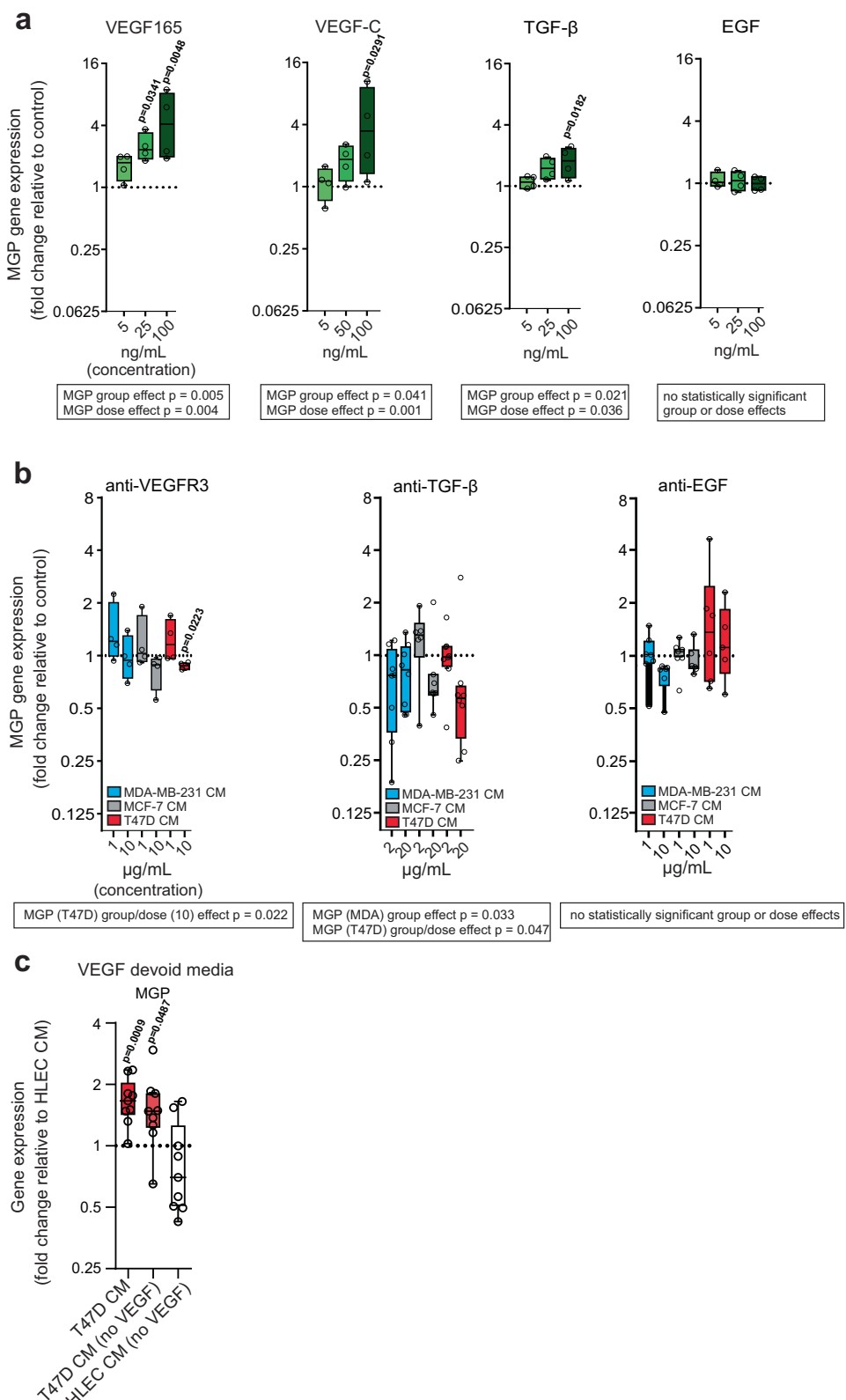

CD200⁺HEY1⁺ LECs in the metastatic lymph node environment. Nonetheless, future fate-mapping studies will be necessary to experimentally validate these transitions.

The increased LEC subset in the paracortical sinuses is located at sites where immune cells leave LNs. Expansion of this subset in metastatic LNs may provide the cancer cells with an improved opportunity to leave the LNs and migrate systemically. The reduced subsets of SCS floor and medullary sinus LECs exhibit high expression

of molecules involved in immunological defenses and inflammation compared with other LN LEC subsets[15]. These sinus LECs produce CSF1 and are crucial for maintaining SCS and medullary sinus macrophages[25], elements vital for bacteria[40] and virus infection[41], as well as tumor immunity[26]. Hence, our study indicates that tumor metastasis within LNs instigates a transition from existing LEC subsets, such as SCS floor LECs, to those carrying markers for immunosuppressive phenotypes.

**Fig. 7 | MGP expression in LN LECs is affected by VEGF and TGF-β. a** Expression of *MGP* in LECs after direct exposure to recombinant VEGF165 (5, 25, or 100 ng/mL), VEGF-C (5, 50, or 100 ng/mL), TGF-β (5, 25, or 100 ng/mL), or EGF (5, 25, or 100 ng/mL) are shown (*n* = 4 HLECs). Data were depicted as Tukey box plots and analyzed using one-way ANOVA linear mixed models (Sidak correction) fitted separately for each parameter with group (recombinant vs control), dose and their interaction as fixed effects. **b** Gene expression changes in modified CM-exposed LECs are shown as determined by qPCR. CM was generated in the presence of antibodies against VEGFR3 (1 or 10 μg/mL, *n* = 4 HLECs), TGF-β (2 or 20 μg/mL, *n* = 8–10 HLECs), and EGF (1 or 10 μg/mL, *n* = 3–5 HLECs), compared to isotype control exposed samples and data were depicted as Tukey box plots showing relative gene changes and

analyzed using a one-way ANOVA linear mixed models (Sidak) fitted separately for each parameter with group (antibody vs control) and dose and their interaction as fixed effects. **c** *MGP* expression determined by qPCR. CM generated with culture media devoid of VEGF supplement was used. Data were shown as Tukey box plots (*n* = 9 HLECs). Data were analyzed using a one-way ANOVA linear mixed model (Sidak). The center line of the box plots represents the median, the box the 25th to 75th percentiles and the whiskers the inner fences. Statistics of group and dose effects are presented within the boxes; significant differences in comparison to the controls (defined as 1) are indicated by *p* values. Source data, non-significant *p* values and detailed experiment and *n*-numbers (biologically independent samples of cultured cells) are provided in the Source Data file.

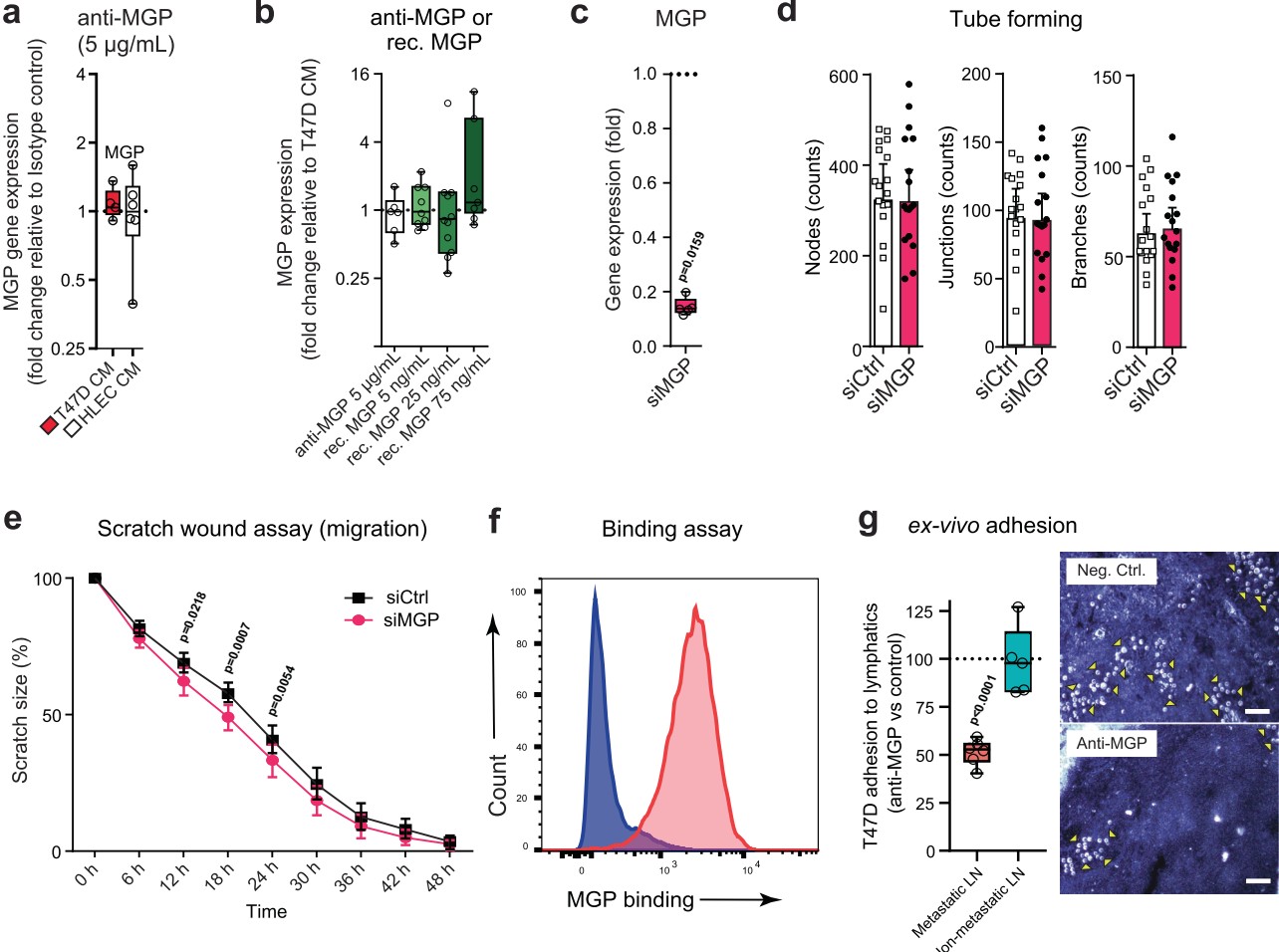

**Fig. 8 | MGP contributes to breast cancer cell adhesion. a** MGP expression in LECs after CM treatment with anti-MGP antibody (*n* = 5–6 HLECs), analyzed using one-way ANOVA linear mixed models (Sidak correction). **b** MGP expression in LECs exposed to anti-MGP antibody or recombinant MGP (*n* = 6–10 HLECs), analyzed using two-way ANOVA linear mixed models (Sidak correction). **c** MGP expression in siRNA-silenced LECs (*n* = 5 HLECs), analyzed by two-sided Mann–Whitney *U*-test. **d** Tube formation quantified by number of nodes, junctions and branches in MGP-silenced vs. control LECs, shown as geometric mean with 95% CI (*n* = 17 HLECs), analyzed with a two-sided paired *t*-test. **e** Scratch assay of MGP-silenced and control LECs over 2 days (mean ± SEM; *n* = 13 HLECs), analyzed by repeated measures two-way ANOVA (matched full mixed model) with Sidak's multiple comparisons. **f** Soluble MGP binding to cancer cells: MGP (His) was added with anti-His antibody

(red), while the control (blue) contained only cancer cells with antibody. Histograms represent two independent experiments. **g** Ex vivo adhesion assay of T47D cells binding to lymphatic sinuses of six metastatic and five non-metastatic LNs from four and five patients, respectively (two-sided ratio paired *t*-test). LN sections were treated with anti-MGP or control antibody. The binding after control antibody was defined as 100% due to day-to-day variation. Example images show T47D cells binding after control vs. anti-MGP treatment to the same metastatic LN area. Adherent cells (some marked by yellow arrowheads) lie on top of tissue sections, focus adjusted to highlight cell adhesion. Scale bar: 50 μm. For box plots: center line = median; box = 25th–75th percentiles; whiskers = inner fences. Source data, non-significant *p* values and detailed experiment and *n*-numbers (biologically independent samples of cultured cells) are provided in the Source Data file.

Lymphatics serve not only as conduits for immune and cancer cell migration but are also active participants in the immune response through antigen presentation. It has been demonstrated that LN LECs express self-antigens and present them on MHC class I and II molecules[21]. Despite the lack of costimulatory molecules, LN LECs do

exhibit high expression of inhibitory molecules such as PD-L1, thereby inducing peripheral tolerance by prompting apoptosis in cytotoxic memory CD8+ T cells[42] and driving PD-1 expression of CD8+ T cells[43]. Our prior research showed that, in particular, SCS floor and medullary sinus LECs strongly express PD-L1 in both mice and humans[15,16]. Given

the selective expression of PD-L1 on LECs, but not other cell types in LNs, the reduction of PD-L1[+] SCS floor LEC subsets and the accumulation of PD-L1[−] LEC subsets in metastatic LNs may significantly impact the efficacy of immune checkpoint therapy, such as anti-PD-L1 agents like atezolizumab and durvalumab, in cancer patients.

Based on the current patient cohort and data, it is impossible to predict, which of the found changes have an impact on clinical outcome. This is due to the relatively small number of patients without sufficiently long follow-up times. However, a recent study reported immunohistochemical analyses of LNs of breast cancer patients and found that high PDPN expression in the lymphatics of sentinel LNs (also found by us, Fig. 3) is a prognostic parameter for worse overall survival, and independent of tumor size, nodal status, and age at the surgery[44].

Similar to the results obtained from the single-cell sequencing of patient samples, we found gene alterations caused by cancer cell CM in in vitro experiments. As we aimed to find universal effects on gene regulation, we focused on changes shared between the different cancer cell lines. When looking at common changes induced by CM, we observed a shift of the endothelial cells to a more anti-inflammatory state, a finding that has previously been found in other cell types such as macrophages and T cells[45–47]. This phenotypical change is therefore likely beneficial for tumor growth and expansion.

MGP was constantly upregulated both in the lymphatics of patients with cancer and in vitro LEC cultures. This is in line with a recent study analyzing changes in cocultures of human melanoma cell lines and LECs using scRNA-seq[48]. In those analyses, *MGP* together with *BGN*, *SOX4*, and *MFAP2*, which are also among our best hits, were significantly upregulated. Although the literature is mostly focused on the role of MGP in calcification and vascular morphogenesis[49,50], it has also been associated with the outcome of various cancers. However, the role of MGP in terms of cancer appears complicated, as not only can its expression be considered both beneficial and detrimental, depending on the cancer type, but also its molecular effect seems to vary[51–55]. Furthermore, MGP expression correlates differently to metastatic spread depending on the tumor type and high expression associates with the metastatic spread in breast cancer[37].

As the role of MGP in LECs is not known, we wanted to focus on its functional importance, especially in LECs. Silencing of MGP increased LEC migration in the scratch assay, suggesting a migration-inhibitory function of MGP. In addition, cancer cell adhesion was decreased in MGP-inhibited LECs and recombinant MGP-bound cancer cells. Such a function could therefore be an indication of a potentially induced facilitation of cancer cell spreading. In fact, Zandueta et al. have reported MGP on osteosarcoma cells to promote their adhesion to and transmigration via endothelial cells, leading to enhanced metastasis[56]. This, together with our results concerning LECs, indicates that MGP can mediate adhesion in different cell types. Although MGP is a secreted K vitamin-dependent protein, it may use its GLA-domain to remain bound to the cell membrane, as GLA-proteins have been shown to do[57]. Comparable to what we have seen in our study regarding lymphatics, MGP has been described as a molecule that suppresses angiogenic sprouting as well as angiogenesis[58], while others have reported conflicting results describing it as an angiogenesis-promoting molecule[59]. It therefore seems to become clear that the function and effects of MGP are dependent on its environment, the cell type, as well as the condition in which it is studied. Hence, MGP exerts diverse roles with regard to the tumor environment and tumor type, and more in-depth research is required to pinpoint its specific role and importance.

When exploring the potential factors enhancing MGP expression, we first chose candidates among the reported factors secreted by cancer cell lines and found that VEGF-A, VEGF-C, and TGF-β upregulate MGP expression in LECs. These findings are in line with an earlier observation, in which TGF-β upregulated MGP on bovine aortic endothelial cells[60]. Results of some previous work not utilizing LECs indicated that MGP expression is upregulated on bone marrow endothelial progenitor cells by PDGF[61], upregulated by TGF-β in embryonic mouse lungs[62], and downregulated in the trabecular meshwork by TGF-β[63]. These findings indicate that a complex regulation pattern for MGP exists and is dependent on the cell type, their density in cell cultures, and treatment regimens[64].

There are limitations in this study. We cannot completely exclude the possibility that the nondraining LNs would be exposed to some tumor-derived soluble factors, which originate from the draining node being in the same chain. Because there are 20 or even more axillary nodes, it seems unlikely that marked amounts of cancer-secreted factors would end up in the distal node. In addition, ethical requirements do not allow more invasive experimental approaches, including the determination of the time course leading to the alterations in the sentinel nodes. Thus, we believe that the changes in the draining LNs are caused by cancer-derived factors, such as those carried by extracellular vesicles via lymphatics and described in the literature[65], and/or the presence of cancer cells via cell-cell contacts.

Another potential limitation of our study is, that once LN LECs are isolated and cultured, they lose certain phenotype markers and no longer accurately represent individual clusters in vivo. However, they can still serve as a tool to analyze the induction and function of various genes or proteins, as we were able to identify several cancer-induced changes that were shared between the HLECs and certain in vivo clusters. In this context, we analyzed the function of MGP, which was highly upregulated both in lymphatics in vivo and HLECs in vitro and were able to discover its adhesive function supporting cancer cell binding to LECs.

In summary, our study demonstrates that LEC cell heterogeneity in LNs is much greater than previously reported. Moreover, we show that breast cancer modifies the lymphatics in sentinel LNs with respect to their phenotype and function. MGP appeared as the top molecule in LECs upregulated by both different types of breast cancers in vivo and cancer cell lines in vitro.

## Methods

### Human samples

At Turku University Hospital, axillary lymph nodes (AXLNs) were obtained from breast cancer patients who underwent mastectomy combined with axillary clearance. Both sentinel nodes and non-metastatic, distant LNs lacking direct drainage from the tumor were collected. The draining nodes were located through preoperative lymphoscintigraphy with 99mTc nano-colloid and, intraoperatively, by injecting Patent Blue into the tumor to highlight the lymphatic pathways. Usually, one to three sentinel nodes were identified based on visible blue staining and the radioactive signal detected with a hand-held gamma counter. The presence of tumor cells in both metastatic and distant LNs was evaluated by histopathological examination and flow cytometric analysis using pan-cytokeratin staining prior to single-cell sequencing as described below. All procedures were performed under license EMTK: 132/2016. Written informed consent was provided by each patient, and all tissue samples were anonymized and processed in line with the ethical standards of the University of Turku, with approval from the Institutional Review Board of Medicolegal Affairs in Helsinki and Turku, Finland.

### LN LEC isolation

Immediately following surgical removal, human LNs were cut into small pieces, and enzymatically digested for 1 h in RPMI supplemented with 0.2 mg/mL collagenase P, 0.8 mg/mL dispase, and 0.1 mg/mL DNase. To verify the presence of tumor cells, a fraction of the resulting single-cell suspension was fixed and permeabilized, and stained with AF488-conjugated anti-pan-cytokeratin antibody (Thermo Fisher Scientific, MA5-18156, 1:100). LNs positive for pan-cytokeratin were

categorized as metastatic, whereas negative ones were classified as distant. CD45[+] immune cells were removed using the EasySep Human CD45 Depletion Kit (Stem Cell Technologies). The remaining suspension was then stained with PE anti-PDPN (BioLegend, 337004, 1:100), AF488 anti-CD45 (BioLegend, 304019, 1:100), APC anti-CD31 (BioLegend, 303115, 1:100), and the LIVE/DEAD Fixable Near-IR Dead Cell Stain Kit (Thermo Fisher Scientific, L10119) for 30 min. Live CD45[−] PDPN[+] CD31[+] LECs were subsequently sorted on an SH800S cell sorter (Sony) in DMEM supplemented with 10% fetal calf serum (FCS).

## scRNA-seq and data preprocessing

Sorted LECs were immediately counted manually and processed in accordance with the 10x Genomics protocols. Single-cell RNA-seq libraries were generated using the Chromium Single Cell 3′ Library and Gel Bead Kits v2 or v3 (10x Genomics, 120237) together with the Chromium Single Cell A Chip Kit, following the manufacturer's recommendations. Briefly, cells were combined with reverse transcription master mix and encapsulated into nanoliter-scale gel bead−in−emulsions (GEMs) via the GemCode technology. The resulting barcoded cDNA was purified and PCR-amplified. During library construction, P5, P7, sample index i7 (10x Genomics, 120262), and R2 (read 2 primer) sequences were incorporated. Sequencing was carried out on Illumina HiSeq3000 or NovaSeq instruments, targeting an average depth of 50,000 reads per cell. Downstream analysis−including demultiplexing, alignment, and quality assessment −was conducted at the Medical Bioinformatics Centre, University of Turku, using the Cell Ranger package (v3.0.1 or v3.1.0, 10x Genomics).

## scRNA-seq data processing and clustering

Preprocessed data were analyzed using Seurat (version 4.3) for graph-based clustering and differential gene expression analysis. For quality control, we filtered out cells with unique feature counts over 6000 or under 200, and cells with more than 12.5% mitochondrial counts. For integration, each dataset was normalized, and 2000 variable genes in each dataset were identified using the "NormalizeData" and "FindVariableFeatures" functions, respectively. Shared highly variable genes across datasets were identified using the

"SelectIntegrationFeatures" function. Integration anchors were identified on the basis of these genes using RPCA[23] as implemented in the "FindIntegrationAnchors" function. The data were then integrated using "IntegrateData()" and scaled using "ScaleData". PCA and uniform manifold approximation and projection (UMAP) dimension reduction with 30 principal components were performed. A nearest-neighbor graph using the 30 dimensions of the PCA reduction was calculated using "FindNeighbors", followed by clustering using "FindClusters" with a resolution of 0.5. The clustering was visualized with UMAP using "DimPlot". A cluster expressing heat-shock proteins such as HSPA1A/B was removed from further analysis, as these genes were shown to be affected by cell dissociation[66]. Another cluster with high expression of mitochondrial genes like *MT-CO3* and *MT-CO1* was also excluded from further analysis. These two clusters appear consistently regardless of LN metastasis and patients. Thus, they are most likely artifacts arising from technical factors. Markers used to phenotype cells in human LNs included PROX1 and FLT4 (LECs), JAM2 (BECs), COL1A1 (stromal cells), PTPRC (leukocytes), KRT19 (cancer cells), and MKI67 (proliferating cells). The PROX1[+] LEC subsets was gated and subclustered by re-identifying integration anchors within datasets as described above.

## Differential expression analysis using scRNA-seq data

Differentially expressed genes (DEGs) between clusters were identified using "FindAllMarkers". The "Min.pct" parameter (minimum percentage of the gene-expressing cells in each cluster) was set to 0.25 (25%), and "thresh.use" (minimum fold change in the gene expression between each cluster and all other clusters) was set to 0.25 (log2FC).

Violin plots and heatmaps were generated using Seurat's "VlnPlot" and "DoHeatmap" functions or Scillus's "plot_heatmap" function. Pathway enrichment analysis of DEGs was performed using Enricher[67]. For pseudobulk DEG analysis, clusters were separated into distant and metastatic LNs and DEGs between these groups were analyzed using "FindAllMarkers".

## Differential abundance testing

The R package Milo (ver.1.8.1) was used for performing differential abundance testing on the k-nearest-neighbor (KNN) graph[24]. The Seurat dataset was converted into a single-cell experiment format, and the x and y coordinates of UMAP were imported from the Seurat dataset. Covariates in the differential abundance testing were "original.ident" (individual sample identity) and "status" (distant or metastatic). Milo analysis was performed using the standard workflow, and the KNN graph was generated from the latent space available from Seurat.

## Trajectory analysis of LECs

We performed a pseudo-time trajectory analysis of LECs using the Monocle3 package[68,69]. Monocle3 projects cells onto a low-dimensional space using UMAP, groups similar cells together using the Louvain algorithm, and then merges adjacent groups and resolves the paths for individual cells[69]. The setting to include a circular path was turned off as we considered LEC differentiation in metastatic LNs to follow a one-directional progression of cell states. Consistent with an earlier study[15], we selected the root node from ACKR4[+] SCS ceiling LECs.

## Spatial transcriptomics using Visium HD

Human fresh tissue samples were frozen and stored in OCT at −80 °C until use. Sections of 10 μm thickness were obtained using a cryostat and processed according to the manufacturer's manual (10x Genomics Visium HD Fresh Frozen Tissue Preparation Handbook, CG000763, revision D). Briefly, tissue sections were fixed with 4% formaldehyde (BP531-25, Thermo Fisher) in PBS and permeabilized in 1% SDS (AM9820, Thermo Fisher), followed by 70% methanol fixation on ice. Tissue sections were blocked at room temperature for 30 minutes in 2% BSA-PBS (126615, Millipore Sigma) with 0.1% Tween-20 (85113, Thermo Fisher) and RNase inhibitor (3335399001, Millipore Sigma), and incubated with PROX1 antibody (R&D, AF2727, 2 μg/ml) and AF647-conjugated pan-cytokeratin (Cell signaling technology, 4528S, 2 μg/ml) for 1 h at room temperature. After PBST (0.4% Tween in PBS) washes, sections were incubated with the secondary antibody AF488 donkey anti-goat IgG (Invitrogen, A32814, 5 μg/ml). The serial section underwent the hematoxylin and eosin (H&E) staining and brightfield imaging with Ocus40 slide scanner (Grundium Ltd, Tampere, Finland). High-resolution images of the IF-stained tissue sections were acquired using a Leica THUNDER widefield fluorescent microscope (Leica Biosystems, Wetzlar, Germany). Immediately after imaging, coverslips were removed, and slides were assembled with the tissue slide cassettes. Gene expression libraries were prepared according to the manufacturer's manual (Visium HD Spatial Gene Expression Reagent Kits User Guide, CG000685, revision B) by using human-specific probes (1000675, 10x Genomics, Pleasanton, CA, USA). The average fragment size of the libraries was determined by using the Agilent 2100 Bioanalyzer (Agilent Technologies, Santa Clara, CA, USA) and the high-sensitivity DNA kit (5067-4626, Agilent). Finally, libraries were pooled at a concentration of 4 nM and sequenced on Illumina NovaSeq X platform (Illumina Inc., San Diego, CA, USA) with the minimum sequencing depth of 700 million read pairs per fully-covered Capture Area, and with following sequencing parameters; Read 1: 43 cycles, i7 Index: ten cycles, i5 Index: ten cycles, Read 2: 50 cycles.

The resulting sequencing data were processed using the Space Ranger HD pipeline (10x Genomics) with the GRCh38 human genome

as a reference. This enabled alignment of gene expression data to spatial coordinates on the tissue sections, guided by the corresponding high-resolution fluorescence images. Analysis of the pre-processed data was performed using Seurat ver 5.2.0.

## Prediction of cell interactions

NicheNet method (nichenetr 2.1.0) was used for analyses of cell-cell communication[30]. NicheNet predicts which ligands from sender cells regulate the expression of target genes in receiver cells by integrating scRNA-seq data with prior knowledge of signaling and gene regulatory networks. In this study, we initially used NicheNet to identify potential communication between combinations of all 21 cell types exhaustively. We then performed a separate analysis, in which CD200[+] HEY1[+] LECs and SCS floor LECs were defined as receivers and lymphocytes, macrophages, plasmablasts, proliferating cells, cancer cells, stromal cells, and BECs were defined as senders. For CD200[+] HEY1[+] LECs, we considered upregulated genes and for SCS floor LECs, downregulated genes as genes of interest. In both analyses, we used the following cut-offs: log fold change of 0.25, fraction of cells expressing the gene of 0.1, maximum number of ligands of 20, maximum number of targets of 100 and ligand-target score of 0.33.

## Analysis of publicly available scRNA-seq datasets

We obtained the dataset for imiquimod-exposed mice directly from the authors[27]. For quality control, we filtered out cells with unique feature counts over 500,000 or under 400,000 and cells with more than 10 % mitochondrial reads. The dataset with Oxazolone immunized mice was downloaded from GEO (GSE145121)[16]. For quality control, we filtered out cells with unique feature counts over 3000 or under 200, and cells with more than 7% mitochondrial counts. Both datasets were analyzed using Seurat (version 5.2.1). For integration, all layers from both datasets were merged together. All samples were normalized, and 2000 variable genes in each sample were identified. The data were subsequently scaled, and PCA was performed. Merged layers were integrated using "IntegrateLayers" function with method "CCAIntegration". A nearest-neighbor graph using the 30 dimensions of the PCA reduction was calculated using "FindNeighbors", followed by clustering using "FindClusters" with a resolution of 0.5. Finally, the UMAP dimension reduction with 30 principal components were performed and visualized. To analyze LEC subsets in human inflamed LNs, a processed scRNA-seq dataset with inflamed human lymph node stromal cells[28] was downloaded from (https://doi.org/10.6084/m9.figshare.26541127.v1), converted to Seurat format using Seurat (v 5.2.1.), and batch effects across samples were removed as we performed in our own dataset. In addition, a single-cell atlas for endothelial cells in breast cancer[29] was downloaded from GEO (GSE155109) and processed using Seurat (version 5.2.1) standard workflow. We used 30 dimensions of the PCA to calculate nearest neighbors and performed clustering at the resolution of one. UMAP dimension reduction with 30 principal components were performed and visualized.

## Immunohistochemical staining

Fresh human LNs were embedded in OCT compound (Sakura) and frozen on dry ice. The LNs were sectioned at a thickness of 6 μm using a cryostat and fixed with acetone at −20 °C for 5 min. The sections were incubated with 10% FCS added to 0.5% BSA in PBS at room temperature for 1 h to prevent nonspecific binding. Then, they were covered with primary antibodies diluted in the same buffer (10% FCS + 0.5% BSA in PBS) and left for overnight incubation at +4 °C. The following primary antibodies were used: goat anti-human PROX1 (R&D Systems AF2727, 2 μg/ml), rabbit anti-human MARCO (Atlas Antibodies, HPA063793, 3 μg/ml), AF488 mouse anti-pan-cytokeratin (Thermo Fisher Scientific, MA5-18156, 1 μg/ml), mouse anti-human MGP (Novus Biologicals, NBP2-45844, 10 μg/ml), and BV421 mouse anti-human CD200 (Biolegend,

329209, 2 μg/ml). Subsequently, the slides were washed and incubated with the following secondary antibodies for 1 h at room temperature: AF488 donkey anti-goat IgG (Thermo Fisher Scientific, A32814, 5 μg/ml), AF647 donkey anti-goat IgG (Thermo Fisher Scientific, A21447, 5 μg/ml), AF555 donkey anti-rabbit IgG (Thermo Fisher Scientific, A32794, 5 μg/ml), and AF647 donkey anti-rabbit IgG (Thermo Fisher Scientific, A32795, 5 μg/ml). After washing with PBS, the sections were mounted with Pro-Long Gold Antifade Mounting medium with DAPI (ThermoFisher Scientific, P36931). Images were captured using an LSM880 confocal microscope (Zeiss) or Stellaris 8 Falcon FLIM microscope (Leica), and analyzed and quantified using ImageJ.

## Cell lines

Human LECs from LNs were obtained from ScienCell (#2500) and CellBiologics (#H6092) or extracted from patient material as described above and were cultured in their respective media (#1001 and H1168, respectively). MDA-MB-231, HCC1954, T47D, and MCF-7 cells were purchased from ATCC and maintained in the laboratory. T47D and MCF-7 cells were cultured in DMEM supplemented with 10% FCS, 100 U/mL penicillin, and 100 μg/mL streptomycin. MDA-MB-231 cells were also cultured in the described DMEM medium, but additionally had 4 mM L-glutamine and MEM non-essential amino acids (Thermo Fisher, 11140050).

## Generation of conditioned media and cell conditioning

Cancer cells or HLECs were plated in their respective media and let to adhere for 1 day before all media was changed to HLEC media. After 1 day, the media was centrifuged at $450 \times g$ for 10 min before adding it to cultured HLECs for 2 days. In 12-well plates, 100,000 cells were plated for generating, and 45,000 cells were plated for receiving the conditioned media. In 24-well plates, these numbers were 50,000 and 23,000, respectively.

## Coculture of HLECs and T47Ds

HLECs and T47Ds were plated in their culture medium on day 0 in small culture flasks or in a 10-cm culture dish, respectively. On d1, medium from T47D cells was collected, centrifuged at $450 \times g$ for 10 min and transferred to HLECs. To certain HLECs, 1/3 of their cell number was added as T47D cells (~340,000 cells). On d3, the cells were permeabilized, stained for CD31 (1:100), podoplanin (1:100) and MGP (1:150) and analyzed with flow cytometry.

## Silencing MGP

siRNA silencing was performed with the Lipofectamine RNAiMAX reagent (#13778075) and Silencer Select siRNA (s8753) or the negative control #1 following the manufacturer's instructions but using only 50% of the recommended reagent amounts (Thermo Fisher Scientific, Espoo, Finland).

## Tube-forming assay

Growth factor-reduced Matrigel (Corning 356231, Espoo, Finland) was plated in a 96-well flat-bottom plate. Once solidified, 10,000 LECs/well were added and cultured for 1 day. For tube-forming assays with CM, cells were exposed to CM for 2 days before performing the assay. CM was also used during the assay duration.

## Binding assay with recombinant MGP

T47D breast cancer cells (50,000 cells) were incubated on ice for 30 min with or without 5 μg/ml recombinant MGP (His) protein (MedChemExpress P702847) and with anti-His-PE antibody (1:25; BioLegend, 362603) in RPMI + 2% FCS. Thereafter, the cells were washed, fixed with 2% PFA for 10 min, washed again with 2% FCS in PBS, and analyzed using flow cytometry (LSRFortessa, BD). All steps were performed on ice and protected from light.

## Adhesion assays

To investigate MGP function at the protein level, ex vivo adhesion assays were performed as described[70]. Briefly, non-metastatic and metastatic LN sections were treated with anti-MGP antibody (1:100, Proteintech 10734-1-AP) or a negative control antibody (Rabbit IgG 1:100, Proteintech 30000-0-A) for 30 min. Following antibody removal, T47D cells were added and incubated in static conditions for 15 min, followed by gentle rotation at 60 rpm for 5 min, and another 15 min in static conditions at 7 °C. The adherent cells were fixed in 1% glutaraldehyde. The number of tumor cells attached to the lymphatic sinusoids was quantified with a dark-field illumination microscope (x200; Leitz Aristoplan, Oberkochen, Germany). To be able to compare experiments performed on different days, the results of these assays are presented as the percentage of control binding, defined as 100%.

## Reagents used in in vitro experiments

Reagents were obtained from different manufacturers. VEGF-C (100-20CD), VEGF165 (100-20), M-CSF, GM-CSF, and G-CSF were obtained from Peprotech, and MGP (TP760483) was obtained from Origene. AACOCF3 (ab120350) was obtained from Abcam, fludarabine (S1491) from Seleckchem, antibodies targeting adrenomedullin (AF6108), EGF (AB-236NA), goat IgG control (AB-108-C), PDGF (AB-20-NA), rabbit IgG control (AB-105-C), and VEGF165 (AB-293-NA) from R&D Systems. Anti-MGP (A5439) was obtained from Abclonal, and anti-TGF-β (BE0057) and the mouse IgG1 isotype (BE0083) from BioXCell. The anti-VEGFR3 antibody (IMC-3C5) was a kind gift from Kari Alitalo, and the control antibody was obtained from Bio-X-Cell (human IgG1, #BE0297). Anti-CD200 antibody (#329209) was obtained from Biolegend (used at 5 µg/mL), the mouse IgG1 isotype from BD (#562438). Anti-CD69 BV605 (used 1:50) and isotype (310937, 400162) were from Biolegend, anti-CD25 BV650 (used 1:50) and isotype (563719, 562652), and anti-CD279 FITC (used 1:50) and isotype (557860, 555748) were from BD.

## Cancer cell proliferation

Proliferation was followed by labeling the cells with 6.5-µM CellTracker Red CMTPX (C34552, Thermo Fisher Scientific) for 40 min and imaged using a Cytation5 (Agilent BioTek) every 7–8 h over a duration of 2 days.

## Co-culture and activation of HLECs and T cells

HLECs (from CellBiologics (#H6092) and cells extracted by ourselves) were plated in a 96-well-flat-bottom plate and grown to confluency. T cells from healthy volunteers were extracted with the EasySep Human T cell Isolation Kit (Stemcell #17951). 30 minutes before T cells were added to the HLEC culture, 5 µg/mL of anti-CD200 blocking antibody or a control antibody (both from Seleckchem, #A2632 and #A3176) were added at 5 µg/mL to the HLECs. After 1 h, stimulator beads (anti-CD3/CD28 Dynabeads (Thermo Fisher Scientific 1131D) and 30 U/mL human IL2 (Miltenyi Biotec, 130-097-744) were added. After 3 days, the cells were analyzed by flow cytometry.

## Explant cultures

CM from explant cultures were collected as described earlier[71]. Briefly, breast cancer tumors and adjacent cancer-free tissues were processed at room temperature within 1 h after surgery. Tissue pieces were cultured in RPMI supplemented with 10% FBS, 1% GlutaMAX, and P/S for 24 h. Thereafter, the CM was aliquoted and frozen at −70 °C until the use. Tumor or adjacent tissue-CM was added at a ratio of 1:10 to HLEC cultures for 48 h.

## Flow cytometric analysis of MGP

Cells were blocked for 20 min with human Ig on ice, permeabilized with the BD Cytofix/Cytoperm Kit (554714), and sequentially stained for 30 min with the following antibodies: MGP (NBP2-45844) from Novus (1:150); negative control, mouse IgG2a (553454) from BD and anti-mouse IgG2a AF546 (A21133) from Invitrogen.

## Scratch wound assays

HLECs were plated on fibronectin-coated wells of a 96-well plate (1800 cells/well), and 1 day later, the cells were siRNA-silenced and cultured for 3 days. A scratch was made to confluent wells with a 200 µL pipette tip, and the well was then imaged with an IncuCyte S3 (Sartorius) every 6 h.

## qPCR

RNA was extracted using the NucleoSpin RNA Kit (Macherey-Nagel, Düren, Germany) according to the manufacturer's instructions. RNA was converted to cDNA using the SuperScript VILO cDNA Synthesis Kit (Thermo Fisher Scientific, Espoo, Finland) before being used in Taq-Man Gene Expression Assays (Applied Biosystems, Stockholm, Sweden) with a QuantStudio3 (Applied Biosystems). To evaluate gene expression levels, the $2^{(-ddCT)}$ method with B2M as a control housekeeping gene was used. The following probes were used:

ACKR4 (Hs00664347_s1), ADORA2A (Hs00169123_m1), ANGPTL4 (Hs01101127_m1), B2M (Hs99999907_m1), CCL20 (Hs00355476_m1), CDH2 (Hs00983056_m1), CDH5 (Hs00901465_m1), CX3CL1 (Hs0171086_m1), H19 (Hs00262142_g1), HLX (Hs00172035_m1), IL4I1 (Hs00541746_m1), MGP (Hs00179899_m1), RAMP3 (Hs00389131_m1), SNAI1 (Hs0019559_m1), SOCS3 (Hs02330328_s1), and VCAM1 (Hs01003372_m1).

## RNA-seq

Total RNA was extracted from LECs using the NucleoSpin RNA Kit (Macherey-Nagel), and library preparation was performed according to the TruSeq Stranded mRNA Sample Preparation protocol using the TruSeq Stranded mRNA HT Kit (Illumina). Sequencing was performed using an Illumina NovaSeq 6000 with a two 50-bp read length at the Finnish Functional Genomics Centre at the University of Turku and Åbo Akademi, as well as Biocenter Finland.

## Bioinformatics for bulk RNA-seq

Raw sequencing data (FASTQ files) were uploaded to the BaseSpace Sequence Hub (Illumina). Quality control was done in BaseSpace (FastQC). This was followed by aligning the sequences against the human reference genome hg19 (UCSC, RefSeq gene annotation) with the RNA-Seq Alignment application (STAR aligner for read mapping and salmon for quantification of reference genes and transcripts). Gene expression differences were identified with the RNA-Seq Differential Expression application (DESeq2). Genes exhibiting a fold change >2 (log2ratio ≥1 and ≤−1) and q values (false discovery rate) <0.05 were considered as differentially expressed genes (DEGs).

Additional analyses were done using the web tools from Bio-Juppies and MetaScape[72]. In particular, the heatmap was generated using Clustergrammer[73], which normalized raw gene counts using the logCPM method, filtered and transformed using the Z-score method.

## Statistics and reproducibility

Data were analyzed using GraphPad Prism 6.02 or 9 (GraphPad Software, San Diego, USA) and SAS version 9.4 (SAS Institute Inc., Cary, NC, USA). The statistical test used is indicated in each figure legend. Values of $*p < 0.05$, $**p < 0.01$, and $***p < 0.001$ were considered statistically significant. A linear mixed model for repeated measurements for dose and group effects was used, and when groups were compared at several dose levels, we applied Sidak's adjustment for multiple comparisons. No data randomization, blinding, sample size estimation, testing of statistical assumptions, or data exclusion was performed. Each replicate shown in the figures corresponds to a biologically distinct sample, i.e., the cells were either from a different donor source or had a different cell passage.

## Reporting summary

Further information on research design is available in the Nature Portfolio Reporting Summary linked to this article.

## Data availability

The scRNA-seq, bulk RNA-seq, and Visium HD spatial transcriptomics datasets generated in this study has been deposited in the GEO database under accession codes GSE248214, GSE248076 and GSE298872, respectively. The scRNA-seq dataset of mouse and human inflamed LNs are available from GEO (GSE145121) [https://www.ncbi.nlm.nih.gov/geo/query/acc.cgi?acc=GSE145121] and the figshare platform [https://doi.org/10.6084/m9.figshare.26541127.v1], respectively. The scRNA-seq dataset of human endothelial cells in breast cancer is available from GEO (GSE155109) [https://www.ncbi.nlm.nih.gov/geo/query/acc.cgi?acc=GSE155109]. Source data are provided with this paper.

## Code availability

Code is available in GitLab at https://gitlab.utu.fi/akitak/lymphatics-in-cancer.

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

## Acknowledgements

We thank Teppo Huttunen from Estimates for statistical analyses, Sari Mäki and Riikka Sjöroos for technical help, Joe Hettinger for revising the language, Bishwa Ghimire for his advice on bioinformatic analyses, Dinghao Zheng for running the Space Ranger pipeline, Damien Kaukonen for running the Space Ranger pipeline and statistical consultation, and Prof. Kari Alitalo for providing us with the anti-VEGFR3 antibody. This work was financed by the Research Council of Finland (A.T., M.H. and S.J.), the Finnish Cancer Foundation (M.H. and S.J.), Sigrid Juselius Foundation (A.T., M.H. and S.J.), Jane and Aatos Erkko Foundation (S.J.) and Sakari Alhopuro Foundation (A.T. and D.E.). The Turku Bioscience Centre Single-cell Omics Core Facility, the Finnish Functional Genomics Centre and Biocenter Finland are acknowledged for infrastructure support.

## Author contributions

D.E., M.H., A.T. and S.J. planned the study. D.E., D.L., M.U., M.L. and A.T. performed the experiments and D.E., D.L., M.U., M.H., A.T. and S.J. analyzed the results. P.B. and I.K. provided the clinical material and pathological analyses. K.E., M.P., T.L., A.K., T.A. and A.T. were responsible for bioinformatics. D.E., A.T. and S.J. prepared the manuscript with contributions from all authors.

## Competing interests

The authors declare no competing interests.
