## [Transparent Peer Review file · Nature Communications]

Breast Cancer Remodels Lymphatics in Sentinel Lymph Nodes

Corresponding Author: Sirpa Jalkanen

Version 0:

Reviewer comments:

Reviewer #1

(Remarks to the Author)

The lymph node stromal cells play a critical role in maintaining the highly organized structure of the node and guiding immune responses against foreign pathogens or tumors. Among these cells, lymph node endothelial cells line the subcapsular sinus, paracortical sinus, and medulla, forming a conduit system within the node. Despite their significant roles in regulating immune responses, little is known about how they respond to cancer metastasis in the node, particularly in humans.

In this study, Eichen et al. characterized the heterogeneity and plasticity of LECs in freshly isolated human breast cancer sentinel lymph nodes and paired distant non-metastatic lymph nodes using single-cell RNA sequencing. This work builds upon their previous study, which identified six distinct populations of LECs in normal human lymph nodes. With advancements in sequencing platforms and bioinformatic analysis pipelines, the current study identified additional LEC clusters in both non-metastatic and metastatic lymph nodes (metLNs). From these data, the authors identified two LEC clusters (clusters 3 and 4) enriched in metLNs and two clusters (clusters 6 and 12) that were reduced in metLNs. Through bioinformatic analysis, they found that Matrix Gla Protein (MGP) was one of the most significantly upregulated genes in LECs from metLNs. Mechanistically, the study demonstrated that TGF- β and VEGFA/C directly enhance MGP expression in LECs. The upregulation of MGP in LECs promotes cancer cell adhesion to the lymphatics, accelerating lymph node metastasis.

Overall, the manuscript is well-written, and the study provides valuable insights into the crosstalk between LECs and cancer cells during breast cancer lymph node metastasis. Below are specific comments to further improve the quality of the manuscript.

Major:

1. The reprogramming of LECs in metLNs might result from their response to cancer cell invasion as they attempt to mount an anti-cancer immune response. It might also be worthwhile to examine the subsets of LECs and their gene signatures in immune-activated LNs as well.
2. How do these reprogramed LECs, for instance the CD200+HEY1+ LECs, impact dendritic cells and T cell interaction and activation in the lymph node?
3. Blocking MGP suppresses cancer cell adhesion to LECs in the ex vivo setting, how about the in vivo setting?
4. Is MGP also expressed in LECs in the primary tumors to enhance cancer cells lymphatic intravasation? Or it is restricted to lymph node metastasis.
5. What are the receptors for MGP on cancer cells? Can they be targeted to prevent the interaction between cancer cells and LECs?

Minor:

1. The discussion is extremely long. Please consolidate the discussion.

2. Matrix Gla Protein is an extracellular matrix glycoprotein, is flow cytometry the proper approach to measure the levels of MGP in LECs?
3. Figure 4d, the lymphocyte-derived cytokines seems missing.
4. Figure 7e, the scratch wound assay shows modest difference, what kind of statistical test was used for the analysis? The standard errors are high makes it hard to believe these are significant.
5. Line 90: please correct the sentence. FRCs should also be PDPN positive.
6. Line 435-436, MGP appears twice in the sentence.
7. Line 316-318, rephrase the sentence in a better way. The current is difficult to understand.

Reviewer #2

(Remarks to the Author)

Eichin et al. characterized changes of LEC transcriptome in human lymph nodes with breast cancer metastases. This is an important study which builds on the previous work that characterized LEC populations in normal lymph nodes by scRNAseq and has identified six distinct types of lymph node LECs. This work expands previous findings of normal lymph nodes, adding to the characterization and examines alterations of LECs when metastases are present. Validation of the transcriptome findings by protein analysis and deeper insight into the key LEC subsets associated with metastases would further strengthen the manuscript.

1. Core of the study is single-cell analysis of LECs. Thorough characterization has been performed and interesting findings presented. However, study would benefit from further validation of the key data by protein analysis. In particular, validation of the key LEC subsets identified as enriched in the LNs with metastases by protein analysis and spatial analysis in the LNs would strengthen the manuscript.
2. Cluster 4 LECs, which are CD200+Hey1+ LECs are significantly increased in LNs with metastases. Further characterization of this subset would be a plus. Confirming existence of this subset by protein analysis, localization in the LNs, proliferation status. Insight into the functional status of the key subsets or the mechanism of how tumor modifies LEC populations would be important to address. It has been reported that CD200+ is immunosuppressive to myeloid cells - does it have this role in LECs?
3. Authors have identified MGP to be upregulated in sentinel LNs of patients with metastases and have performed initial in vitro analyses to interrogate its function. Further in depth analyses should be performed. MGP is a vitamin K dependent protein that is expressed in many tissues, such as bone, cartilage, heart, kidney. How unique is MGP expression to LECs? What are known functions of MGP that could affect tumor cells? Is MGP expressed in other cells in the lymph nodes? In vitro assays showed that MGP inhibits tumor cell adhesion to LECs, however, further studies are needed to establish the importance of increased adhesion, if any, for metastasis. This cannot be assumed.
4. In vitro studies examined the effects of tumor cell CM on LECs, however, it has not been proven that the observed changes are specific to tumor cells. Normal breast epithelial cells should be used as a control. LECs may change also in co-culture with normal epithelial cells.
5. Authors have indicated that LECs transition from immunologically active LEC subsets to those that are immunosuppressive. This is a strong statement that is based in transcriptomic data and needs to be backed up with direct evidence or the statements should be softened. Anti-inflammatory state of LECs may be induced because of a co-culture with (normal) epithelial cells, and may not be specific response to a tumor. Anti-inflammatory state is an overstatement in view of the data. presented, and would have to be better defined.
6. On page 11, it is stated that "Despite existing beliefs that tumors induce proliferation of LECs in murine models, we observed no difference in cell proliferation upon incubation with various media" (Suppl. Fig. 8). This statement needs to be corrected - not all tumor cells will induce lymphangiogenesis and it should not be expected that all tumor cells induce LEC proliferation in vitro. The text should be corrected to indicate that "the cell lines tested did not induce LEC proliferation in vitro under such and such conditions".
7. Discussion includes many hypotheses that are not supported by the data. It should be revised to be less vague and closer to the data. There is a lot of hypotheses about the function of MGP. There are vague statements such as "This may be due to multiple factors in the CM causing different effects"; "We propose that cancer-promoting MGP expression in LECs has the potential to contribute to cancer behavior". Discussion should be streamlined.
8. Discussion: There is a statement that in cell culture, LN LECs dedifferentiate and therefore do not accurately represent different in vivo clusters". That statement is too strong, while LECs may change their properties in the in vitro environment, they maintain lineage characteristics and do not dedifferentiate. This should be rephrased.

Reviewer #3

(Remarks to the Author)

The authors demonstrate a significant shift in LEC composition and phenotype in response to extracellular components produced by breast cancer cells following colonization by metastatic cells - potentially by TGFbeta, VEGFA, and/or VEGFC which have well established roles in tumour microenvironment remodelling. Notably, this shift is associated with reduced inflammation / immunological functions which likely contributes to the immune-escape of metastatic cells. They present a large cohort of scRNAseq from patient-matched tumour-proximal metastatic lymph nodes and distal uncolonized lymph nodes, which revealed a metastatic-specific population of CD200+ HAY1+ LECs possibly driven by response to TGFb expression. In addition, they validate this finding by exposing in vitro LECs to media conditioned by breast cancer cell-lines using flow, qPCR and bulkRNAseq. They further show this shift (as characterized by upregulation of MGP) is inducible by TGFb, VEGFA and/or VEGFC and is associated with greater adherence of tumour cells to LECs supporting its role in colonization of lymph nodes by metastatic cancer cells. Overall I find the results convincing and comprehensive, and revealing a novel specific mechanism of tumour progression.

Major Concerns

The authors state that TGFBR2/TGFBR3 etc.. are expressed on CD200+LECs but the only plot (Figure 4f) provided to support this is of the prior weights used by NicheNet which would not account for the expression in CD200+LECs, could the authors provide direct evidence that these receptors are expressed in the CD200+LECs?

The authors suggest that CD200+ LECs were near metastatic cancer cells, however in the provided figure (Figure 2g) they appear generally ubiquitously present both near and distant from the tumour cells. How did the authors quantify the proximity between different LEC populations to the metastatic cells? In addition, I had difficulty spotting the keratin+ tumour cells indicated by the white arrows.

The authors state that VEGFA is highly expressed by macrophages, but in Figure 4e it is shown that cancer cells express VEGFA at a similar level.

For the trajectory analysis, what is the biological justification for ACK4+SCS ceiling LECs being the root of the trajectory?

Minor Concerns

In figure 1b what does AXLN1-4 mean?

In figure 2e the authors postulate a developmental/differentiation relationship between the different LEC populations. What evidence is there for this? Monocle and most other pseudotime methods simply identify gradients in expression, those gradients do not necessarily represent developmental/differentiation relationships; they can equally represent spatial gradients arising from oxygen or other nutrient availability or tissue organization factors. CCA integration tends to generate gradient-like structures in its lower-dimensional embedding, is there orthogonal evidence that LECs can transition between the identified cell-states?

The authors state in their discussion that updating to Seurat v4 would offer significant improvement in integration over using CCA, but they state in their methods that they are using CCA integration in Seurat v4. They should clarify these contradicting statements.

The authors exclude 2 clusters due to concerns with data quality, while these are well justified, were these clusters associated with specific samples or metastatic or distant LNs? Tumours may trigger stress responses in tissues thus the poor quality of those clusters may not be due to technical artefact, but rather be a real biological effect of tumours on LECs.

For the Monocle analysis does enabling the circular path option have a significant effect on the results? It appears that T2 and T6 could potentially be a circular (i.e. converging) trajectory. What is the justification for disallowing this option?

The authors should specify the version of NicheNet they are using, this package has been revised multiple times since publication, thus the version number is necessary to ensure reproducibility of results.

Which methodology was used for pseudobulk DE for the scRNAseq data?

Reviewer #4

(Remarks to the Author)

In this paper, Eichin et al. used single cell RNA sequence analysis to profile lymphatic endothelial cell (LEC) subsets in paired metastatic and non metastatic lymph nodes (LN) from 9 patients with luminal and HER2+ breast cancer. They found new LEC subsets in metastatic LNs and transcriptional changes. Matrix Gla protein (MGP) was one of the most upregulated genes in all LEC subsets across all patients. MGP expression in LECs was TGFb and VEGF dependent and regulated HLEC migration and cancer cell adhesion to LN lymphatics.

Overall, this is an interesting and very informative study that provides new knowledge on the transcriptional program modulated in human metastatic LN compared to non metastatic LN in breast cancer patients. The high quality data are solid,

convincing and well performed with appropriate controls and statistical analyses. However, the underlying mechanisms and the role of MGP in breast cancer metastasis to LN still remain poorly defined as it is in the present manuscript.

Major comments:

- 1) Figure 5: an HER2+ cell line (HCC1954, SKBR3) should be added to reflect the patients with luminal and HER2+ breast cancer. The rationale to use MDA-MB-231 cell line is not detailed in the manuscript.
- 2) Figure 6: the rationale to test MGP as one of the most upregulated gene by the CM and in the patient LN is clear, however, it would add to test other genes pro-inflammatory (MGP, SOCS3, RAMP3) and anti-inflammatory (VCAM1, IL4) to support the hypothesis of a switch in LECs from a pro-inflammatory to anti-inflammatory state and the pathway analysis in Figure 5e.
- 3) Figure 6b: what about the MCF-7 CM? it seems to be also reduced by the blocking antibodies.
- 4) Figure 7e: the difference in HLECs migration between control siRNA and siMGP is very little and therefore, not very convincing to support a role of MGP in HLECs migration. Since figure 5 and 6 show the increased expression of MGP in response to breast cancer CM and VEGF/TGFb, it would be better to assess cell migration and invasion (boyden chamber assays), and adhesion assays in response to either breast cancer cell lines CM or with VEGF or TGF beta, with control siCtrl, siMGP, or anti-MGP antibodies.

These additional experiments would help strengthen the proposed mechanistic model and the role of TGFb and VEGF-induced upregulation of MGP on breast cancer cell dissemination in patients, which could lead to combined therapeutic strategies for breast cancer patients.

Minor comments:

Figure 1b: not sure what is AXLN1, 2, 3, 4.

Version 1:

Reviewer comments:

Reviewer #1

(Remarks to the Author)

With the additional bioinformatics analysis, spatial transcriptomic data, and the new functional experiments, the authors addressed most of my questions and concerns, and the revised manuscript has been significantly improved.

I am convinced that the CD200+HEY1+ capillary-like LECs are enriched in metastatic LNs compared to distant non-metastatic LNs. Mechanistically, the authors propose the presence of MGP in this new population of LECs promotes cancer cell invasion in the LNs. However, in the Supplemental table 1 Excel file containing the up and down regulated genes in different clusters, I could not find the MGP gene in any of the clusters. And why all the avg_log2FC are negative? What are these upregulated genes? In figure 4 panel C, MGP is broadly expressed in many subclusters of LECs in both non-metastatic distant LNs and metastatic LNs as shown in the violin plot. That means MGP is not only restricted to the capillary-like CD200+HEY1+ LECs (cluster 4). For instance, in the violin plot, MGP is substantially increased in cluster 5 LECs in metastatic LNs as well. Any comments on the other MGP+ clusters of LECs? Furthermore, I am wondering, what other cell types in the metastatic LN also expressed MGP? Could the author check the MGP gene expression in the single-cell datasets?

The new spatial transcriptomic data in figure 3 is very cool, it clearly showed the difference between LECs in distant LNs and metastatic LNs. Is it possible to include MGP in the dotplot in panel D? Also, in the metastatic LN from patient #3, it seems like many of the cells in "Cancer cells and LEC1" cluster are enriched in the SCS, any comments on this population? Was this LN at the very early stage of cancer metastasis that's why many of the cancer cells were enriched in the SCS? It seems like the spatial distribution of LECs in metLNs in Pt#3 and Pt#5 are very different.

Some minor comments:

Figure 1d and Figure 2a are a little bit redundant, consider move one of them to the supplemental figures. In figure 2a the left panel, there are very few cells from cluster #4, but the label of cluster #4 is missing.

In Figure 6c the heatmap, are these little triangles on the right side and the bottom have some specific meaning? Or maybe some labels are missing?

Reviewer #2

(Remarks to the Author)

Eichin et al. addressed most of the concerns raised. Several issues remain to be addressed, mainly related to data presentation and interpretation.

1. Core of the study is single-cell analysis of LECs. Thorough characterization has been performed and interesting findings presented. However, study would benefit from further validation of the key data by protein analysis. In particular, validation of the key LEC subsets identified as enriched in the LNs with metastases by protein analysis and spatial analysis in the LNs would strengthen the manuscript.

AUTHOR'S RESPONSE: To further validate key LEC subsets in their spatial context, we performed high-resolution spatial transcriptomics using Visium HD (Figure 3). This analysis confirmed the marked remodeling of LECs in metastatic LNs, including the reduction of SCS floor and medullary sinus LECs, and the enrichment of BGN+ capillary-like LECs. These reprogrammed LEC subsets were spatially localized to the paracortical and subcapsular sinus regions. Although protein-level validation remains technically challenging due to limited availability of good antibodies for each target, our high-resolution spatial transcriptomics data provides strong evidence supporting the presence, remodeling, and anatomical localization of LEC subsets in situ. This description has been added to the revised manuscript (Figure 3, lines 186-214).

RE-REVIEW: The inclusion of Visium HD data further strengthens the study. In the Results section (lines 186-214), please indicate specifically how the Visium HD data align and do not align with the key scRNA-seq data.

It should be mentioned in the Results that metastases in the LNs were large and that this markedly altered architecture of the entire lymph node. The assumption of the reader is that the lymph node architecture is preserved in presence of metastases, which it can be when metastases are smaller. Clarifying that metastases are large and that LN architecture is markedly changed is important for data interpretation.

2. Cluster 4 LECs, which are CD200+Hey1+ LECs are significantly increased in LNs with metastases. Further characterization of this subset would be a plus. Confirming existence of this subset by protein analysis, localization in the LNs, proliferation status. Insight into the functional status of the key subsets or the mechanism of how tumor modifies LEC populations would be important to address. It has been reported that CD200+ is immunosuppressive to myeloid cells - does it have this role in LECs?

AUTHOR'S RESPONSE: Localization of CD200+ LECs was shown with immunofluorescence in Fig. 2g. To address their function role, we have now performed T cell activation assays in the presence of CD200 expressing HLECs and anti-CD200 antibody. Indeed, blocking CD200 led to an increase in T cell activation, which indicate its potential role in T cell suppression (Supplemental Figure 5, lines 181-185).

RE-REVIEW: Please show evidence that LECs used in this experiment were CD200+ (if shown already, indicate in the Results where has it been shown). In the corresponding Methods section, clarify which LECs specifically were used in the experiments with T cells. Commercially obtained or isolated by your lab, and from what source. Cell lines or primary cells. Several sources are indicated in the methods at the beginning.

3. Authors have identified MGP to be upregulated in sentinel LNs of patients with metastases and have performed initial in vitro analyses to interrogate its function. Further in depth analyses should be performed. MGP is a vitamin K dependent protein that is expressed in many tissues, such as bone, cartilage, heart, kidney. How unique is MGP expression to LECs? What are known functions of MGP that could affect tumor cells? Is MGP expressed in other cells in the lymph nodes? In vitro assays showed that MGP inhibits tumor cell adhesion to LECs, however, further studies are needed to establish the importance of increased adhesion, if any, for metastasis. This cannot be assumed.

AUTHOR'S RESPONSE: LEC expression is not unique to lymphatics but its constant upregulation in metastatic LNs of every single patient is noteworthy. As it is in numerous examples in the literature, studies in many areas have focused on limited aspects of identified molecules and that is reflected in names of many molecules. Importantly, further studies have shown additional properties for those molecules. In case of MGP nothing is known about its role in lymphatics. Thus, its role in adhesion reported in this manuscript is completely new information.

Upregulation of MGP in sentinel nodes concerned lymphatics as despite expressed by the LN capsule, its expression in the capsule was not changed upon metastasis. Moreover, we conducted ex vivo adhesion assays also using distal, non-cancerous LNs. In these assays, anti-MGP antibody did not reduce binding being on line with its low expression in normal LNs. Importantly, we now demonstrate that recombinant MGP binds cancer cells. These results are now presented in Fig 8f and g. These new experiments show that MGP directly supports cancer cell binding and this takes place especially in metastatic LNs.

(For ethical reasons, patient studies are not currently possible. Thus, these ex vivo studies are closest, which are currently possible. These types of ex vivo assays have been highly accurate to reflect what happens in vivo concerning the leukocyte and cancer cell adhesion to vasculature. Alpha 4 integrins targeted by natalizumab and vedolizumab, both well selling drugs on market and inhibiting __ and __ binding to VCAM-1 and MAdCAM-1) are excellent examples, the functions of which were found by this assay method (Yednock et al. Nature 356, 63-66, 1992, Fig. 2 and Hu et al. PNAS 89: 8254-8258, 1992, Fig. 5). Examples on lymphatics are Clever-1 and macrophage mannose receptor, both of which mediate binding in ex vivo assays and are involved in trafficking within the lymphatics in vivo (Irljala et al. J Exp Med 194:1033-1041, 2001, Fig 2; Salmi et al, Circ Res 112:1577-1582, 2013, Fig. 3; Irljala et al. Cancer Res, 63: 4671-4676, 2003, Fig. 1 and Hollmén et al, BJC 123:501-509, 2020: anti-Clever-1 antibody is in clinical trials)).

RE-REVIEW: Addressed

4. In vitro studies examined the effects of tumor cell CM on LECs, however, it has not been proven that the observed changes are specific to tumor cells. Normal breast epithelial cells should be used as a control. LECs may change also in co-culture with normal epithelial cells.

AUTHOR'S RESPONSE: To address this point, we have now conducted additional experiments using CM from tumor tissue and adjacent normal tissue explants from breast cancer patients. Although the observed changes were generally modest, the expression trend for our main gene of interest, MGP, was consistent with that seen in the original experiments. Some of the observed differences are likely attributable to the distinct secretion profiles of tissue explants compared to established cancer cell lines, supporting the interpretation that the originally observed changes are largely tumor cell-specific (Supplemental Figure 13).

RE-REVIEW: Studies shown in Fig. 6 were done using breast cancer cell lines. Primary breast epithelial cells or non-tumorigenic breast epithelial cell line would have been the most relevant control for this experiment. New data shown in Suppl. Fig. 13 using CM from tissue samples show very different results from what is shown in Fig. 6 using cell lines. Data shown in Fig. 6f has not been replicated with the new data in Suppl. Fig. 13. However, this is expected, since the tissues are complex and are expected to have a very different secretome than cultured cell lines, which authors acknowledge.

However, the claim that "the originally observed changes are largely tumor cell-specific" is misleading, because it suggests that non-tumorigenic cells do not induce such changes, and that has not been demonstrated. If the authors want to determine what gene expression changes in LECs are specifically induced by tumor cells, then non-tumorigenic breast epithelial cells should be used as a control. Alternatively, if the data remains as it is now, interpretation needs to be modified - it cannot be stated that the changes are tumor-specific. Factors produced by tumor cell lines induced these changes in LECs, but this does not mean that they are tumor-specific, or that the same will be seen in vivo, as evidenced by the recent experiment.

Suppl. Fig. 13, n=6 means replicates from the same sample for PCR (technical) or CM from 6 different LN samples?

In Fig. 6, please indicate in the Figure legend the time - how long were LECs incubated with tumor CM (48hr?) and for how long was media conditioned (24hr?). It does not matter on which days, but the duration.

5. Authors have indicated that LECs transition from immunologically active LEC subsets to those that are immunosuppressive. This is a strong statement that is based in transcriptomic data and needs to be backed up with direct evidence or the statements should be softened. Anti-inflammatory state of LECs may be induced because of a co-culture with (normal) epithelial cells, and may not be specific response to a tumor. Anti-inflammatory state is an overstatement in view of the data presented and would have to be better defined.

AUTHOR'S RESPONSE: We have removed the part discussing this aspect.

RE-REVIEW: Addressed

6. On page 11, it is stated that "Despite existing beliefs that tumors induce proliferation of LECs in murine models, we observed no difference in cell proliferation upon incubation with various media" (Suppl. Fig. 8). This statement needs to be corrected - not all tumor cells will induce lymphangiogenesis and it should not be expected that all tumor cells induce LEC proliferation in vitro. The text should be corrected to indicate that "the cell lines tested did not induce LEC proliferation in vitro under such and such conditions".

AUTHOR'S RESPONSE: This has been modified as suggested.

RE-REVIEW: Addressed

7. Discussion includes many hypotheses that are not supported by the data. It should be revised to be less vague and closer to the data. There is a lot of hypotheses about the function of MGP. There are vague statements such as "This may be due to multiple factors in the CM causing different effects"; "We propose that cancer-promoting MGP expression in LECs has the potential to contribute to cancer behavior". Discussion should be streamlined.

AUTHOR'S RESPONSE: We have removed these sentences. In addition, we have modified the section discussing CD200 and MGP, and shortened the discussion.

RE-REVIEW: Discussion, Lines 410-411: "MGP silencing/inhibition increased LECs migration and reduced cancer cell adhesion" change to be more clear. Suggestion: "MGP silencing or inhibition increased LECs migration and reduced cancer cell adhesion, respectively."

Discussion, Lines 441-443: "Expansion of this subset in metastatic LNs provides the cancer cells with an improved opportunity to. leave the LNs and migrate systemically" There is no evidence that this subset supports egress of tumor cells, this statement should be softened. Suggestion: "...may provide with an opportunity"

Discussion, Lines 447-450: "Hence, our study indicates that tumor metastasis within LNs instigates a transition from immunologically active LEC subsets....to those that are immunosuppressive and conducive to tumor metastasis". This is an oversimplification, please reformulate. "Immunologically active". Previous studies have shown that LECs in lymph nodes promote peripheral tolerance; studies by Engelhard for example have shown complexities of lymph node LEC contribution to immunity.

8. Discussion: There is a statement that in cell culture, LN LECs dedifferentiate and therefore do not accurately represent different in vivo clusters". That statement is too strong, while LECs may change their properties in the in vitro environment, they maintain lineage characteristics and do not dedifferentiate. This should be rephrased.

AUTHOR'S RESPONSE: The referee is correct that certain lymphatic markers remain, while many phenotype markers are lost during the culture, especially in higher passages. The sentence has been modified (Lines 522-523).

RE-REVIEW: Addressed

Reviewer #3

(Remarks to the Author)

I thank the authors for their thorough explanations, and the manuscript is significantly improved and should be accepted for publication.

Reviewer #4

(Remarks to the Author)

The authors have performed new experiments/analyses and have added new convincing data that satisfactorily addressed the comments' of the reviewers. This is a study of great interest and significance in the field of cancer research.

Minor corrections:

Line 294: HER2+

Line 374: Fig. 8a

Version 2:

Reviewer comments:

Reviewer #1

(Remarks to the Author)

I thank the authors for addressing all my questions and concerns. The additional analyses including the spatial data of pt#3, the excel spreadsheet of DEGs, as well as the figure legends and discussion are all satisfied. The revised manuscript should be accepted for publication.

Reviewer #2

(Remarks to the Author)

Authors have addressed all concerns raised. This study sheds light on important tumor-induced changes in the LN and inspires further lines of investigation. Congratulations on your work!

June 10th, 2025

Dear Editor

Please find our revised manuscript attached. We sincerely thank the reviewers for their constructive feedback, which has significantly helped us to improve the quality of the manuscript. We have carefully addressed all the comments and concerns raised. A detailed point-by-point response is provided below, and all revisions made to the manuscript are marked in red. We hope that the revised version is suitable for publication.

Best regards

Sirpa Jalkanen, MD, PhD

Professor

REVIEWER COMMENTS

Reviewer #1 (Remarks to the Author):

The lymph node stromal cells play a critical role in maintaining the highly organized structure of the node and guiding immune responses against foreign pathogens or tumors. Among these cells, lymph node endothelial cells line the subcapsular sinus, paracortical sinus, and medulla, forming a conduit system within the node. Despite their significant roles in regulating immune responses, little is known about how they respond to cancer metastasis in the node, particularly in humans.

In this study, Eichen et al. characterized the heterogeneity and plasticity of LECs in freshly isolated human breast cancer sentinel lymph nodes and paired distant non-metastatic lymph nodes using single-cell RNA sequencing. This work builds upon their previous study, which identified six distinct populations of LECs in normal human lymph nodes. With advancements in sequencing platforms and bioinformatic analysis pipelines, the current study identified additional LEC clusters in both non-metastatic and metastatic lymph nodes (metLNs). From these data, the authors identified two LEC clusters (clusters 3 and 4) enriched in metLNs and two clusters (clusters 6 and 12) that were reduced in metLNs. Through bioinformatic analysis, they found that Matrix Gla Protein (MGP) was one of the most significantly upregulated genes in LECs from metLNs. Mechanistically, the study demonstrated that TGF- β and VEGFA/C directly enhance MGP expression in LECs. The upregulation of MGP in LECs promotes cancer cell adhesion to the lymphatics, accelerating lymph node metastasis.

Overall, the manuscript is well-written, and the study provides valuable insights into the crosstalk between LECs and cancer cells during breast cancer lymph node metastasis. Below are specific comments to further improve the quality of the manuscript.

Major:

1. The reprogramming of LECs in metLNs might result from their response to cancer cell invasion as they attempt to mount an anti-cancer immune response. It might also be worthwhile to examine the subsets of LECs and their gene signatures in immune-activated LNs as well.

Unfortunately, we do not have permission to collect immune-activated lymph nodes (LNs) due to ethical reasons. Normally, lymph nodes are biopsied only when cancer is suspected, and in positive cases, they are removed (together with several surrounding lymph nodes, as in breast cancer) as part of cancer surgery. It is also impossible for us to obtain both an inflamed and a normal lymph node from the same individual, which would be necessary to directly compare them and draw conclusions about the effects of inflammation.

Instead of generating a new dataset, we reanalyzed a recently published dataset of human inflamed LNs (Lütge M et al., *Sci Immunol* 2025). LEC subsets were extracted and re-clustered following batch correction across samples. No significant alterations in LEC subset composition were observed in inflamed LNs. Additionally, we analyzed publicly available mouse datasets in which mice were immunized with oxazolone or imiquimod (Xiang M et al., *Front Cardiovasc Med* 2020; Sibling E et al., *Cells* 2021). Similarly, the LEC subset composition remained largely unchanged in murine inflamed LNs. Genes upregulated in human metastatic LNs, such as MGP and BGN, were not upregulated in the inflamed LNs. These findings suggest that LEC remodeling observed in metastatic LNs was not primarily driven by inflammation. This analysis and a new figure have been added (Lines 243-249 and Supplemental Figure 8).

2. How do these reprogrammed LECs, for instance the CD200+HEY1+ LECs, impact dendritic cells and T cell interaction and activation in the lymph node?

To address this question, we co-cultured CD200-expressing LECs with T cells in the presence or absence of anti-CD200. The interactions between LECs and T cells reduced the expression of T cell activation markers such as CD25 and CD69 in a CD200-dependent manner, indicating the reprogrammed LECs are indeed immunosuppressive. These additions are in Lines 181-185 and Supplemental Figure 5.

3. Blocking MGP suppresses cancer cell adhesion to LECs in the ex vivo setting, how about the in vivo setting?

For ethical reasons, we are unfortunately not yet able to test this in patients. Thus, the *ex vivo* studies currently represent the closest feasible approximation to the clinical situation. These types of *ex vivo* assays have proven highly accurate in reflecting *in vivo* behavior, particularly regarding leukocyte and cancer cell adhesion to the vasculature. Alpha-4 integrins, targeted by the widely used drugs natalizumab and vedolizumab—which inhibit $\alpha 4\beta 1$ and $\alpha 4\beta 7$ binding to VCAM-1 and MAdCAM-1, respectively, serve as excellent examples. The functions of these integrins were originally identified using this assay method (Yednock et al., *Nature* 356: 63–66, 1992, Fig. 2; Hu et al., *PNAS* 89: 8254–8258, 1992, Fig. 5). In the context of lymphatics, Clever-1 and the macrophage mannose receptor are additional examples. Both mediate binding in *ex vivo* assays and are involved in cell trafficking within the lymphatics *in vivo* (Irjala et al., *J Exp Med* 194: 1033–1041, 2001, Fig. 2; Salmi et al., *Circ Res* 112: 1577–1582, 2013, Fig. 3; Irjala et al., *Cancer Res* 63: 4671–4676, 2003, Fig. 1; Hollmén et al., *BJC* 123: 501–509, 2020). Notably, an anti-Clever-1 antibody is currently in clinical trials.

4. Is MGP also expressed in LECs in the primary tumors to enhance cancer cells lymphatic intravasation? Or it is restricted to lymph node metastasis.

To address this question, we analyzed a publicly available scRNA-seq dataset of endothelial cells from human breast tumors (Geldhof V et al., *Nat Comm*, 2022). This analysis revealed that MGP expression was elevated in LECs within primary tumors compared to those in peritumoral regions. To validate this observation at the protein level, we performed immunostaining of human breast tumor tissues and adjacent healthy regions with an anti-MGP antibody. Consistently, MGP was upregulated in LECs within tumor regions. This description and the corresponding data were added to the revised manuscript (Lines 249-254 and Supplemental Figure 9).

5. What are the receptors for MGP on cancer cells? Can they be targeted to prevent the interaction between cancer cells and LECs?

The counter-receptor of MGP on cancer cells is not yet known. However, by using recombinant human MGP we now show that it directly binds cancer cells. The new results have been added, Fig 8f (Line 392-394).

Minor:

1. The discussion is extremely long. Please consolidate the discussion.

The discussion has been shortened.

2. Matrix Gla Protein is an extracellular matrix glycoprotein, is flow cytometry the proper approach to measure the levels of MGP in LECs?

The referee is correct that MGP is mainly presented as an extracellular matrix glycoprotein, but it has the structure (GLA domain) allowing it to remain bound to the cell surface (as written, Lines). Small amounts of it (red histograms below) are indeed cell bound as shown in this histogram below (Figure 1).

Figure 1. Immunofluorescence analyses of MGP expression on HLECS with and without VEGF-C and TGF- β stimulation

3. *Figure 4d, the lymphocyte-derived cytokines seems missing.*

This has been corrected.

4. *Figure 7e, the scratch wound assay shows modest difference, what kind of statistical test was used for the analysis? The standard errors are high makes it hard to believe these are significant.*

Scratch assay of MGP-silenced and control LECs was performed over the course of 2 days, resulting in longitudinal data. Data were shown as mean with 95% CI (n = 13) and were analyzed using repeated measures two-way ANOVA (matched fitted full mixed model) and the p-values were adjusted by Sidak's multiple comparison test. We have now shown these data in Figure 8e, presented with mean \pm SEM, which better corresponds to statistical significance testing. We would like to note, that repeated measures ANOVA accounts for dynamic changes in longitudinal data, allowing for the detection of significant differences beyond fixed time points with greater statistical power.

5. *Line 90: please correct the sentence. FRCs should also be PDPN positive.*

We apologize for this confusion. In this line we were not meaning to say that they are negative, we just wrote that PDPN expression was not increased in FRC of metastatic LNs. To avoid such confusion, we modified the sentences (Lines 90-92):

'PDPN protein expression on LECs was higher than that in distant LNs, indicating that tumor metastasis alters LECs in metastatic LNs. No increase was seen on fibroblastic reticular cells (FRCs) or blood endothelial cells (BECs) of metastatic LNs'.

6. *Line 435-436, MGP appears twice in the sentence.*

Extra MGP depleted.

7. *Line 316-318, rephrase the sentence in a better way. The current is difficult to understand.*

Rephrased:

VEGFs are responsible for the effect as the culture medium without VEGF did not cause the effects (Lines 364-365).

Reviewer #2 (Remarks to the Author):

Eichin et al. characterized changes of LEC transcriptome in human lymph nodes with breast

cancer metastases. This is an important study which builds on the previous work that characterized LEC populations in normal lymph nodes by scRNAseq and has identified six distinct types of lymph node LECs. This work expands previous findings of normal lymph nodes, adding to the characterization and examines alterations of LECs when metastases are present. Validation of the transcriptome findings by protein analysis and deeper insight into the key LEC subsets associated with metastases would further strengthen the manuscript.

We have now performed spatial transcriptomics and additional stainings as explained below.

1. Core of the study is single-cell analysis of LECs. Thorough characterization has been performed and interesting findings presented. However, study would benefit from further validation of the key data by protein analysis. In particular, validation of the key LEC subsets identified as enriched in the LNs with metastases by protein analysis and spatial analysis in the LNs would strengthen the manuscript.

To further validate key LEC subsets in their spatial context, we performed high-resolution spatial transcriptomics using Visium HD (Figure 3). This analysis confirmed the marked remodeling of LECs in metastatic LNs, including the reduction of SCS floor and medullary sinus LECs, and the enrichment of BGN⁺ capillary-like LECs. These reprogrammed LEC subsets were spatially localized to the paracortical and subcapsular sinus regions. Although protein-level validation remains technically challenging due to limited availability of good antibodies for each target, our high-resolution spatial transcriptomics data provides strong evidence supporting the presence, remodeling, and anatomical localization of LEC subsets in situ. This description has been added to the revised manuscript (Figure 3, lines 186-214).

2. Cluster 4 LECs, which are CD200+Hey1+ LECs are significantly increased in LNs with metastases. Further characterization of this subset would be a plus. Confirming existence of this subset by protein analysis, localization in the LNs, proliferation status. Insight into the functional status of the key subsets or the mechanism of how tumor modifies LEC populations would be important to address. It has been reported that CD200+ is immunosuppressive to myeloid cells - does it have this role in LECs?

Localization of CD200⁺ LECs was shown with immunofluorescence in Fig. 2g. To address their function role, we have now performed T cell activation assays in the presence of CD200 expressing HLECs and anti-CD200 antibody. Indeed, blocking CD200 led to an increase in T cell activation, which indicate its potential role in T cell suppression (Supplemental Figure 5, lines 181-185).

3. Authors have identified MGP to be upregulated in sentinel LNs of patients with metastases and have performed initial in vitro analyses to interrogate its function. Further in depth analyses should be performed. MGP is a vitamin K dependent protein that is expressed in many tissues, such as bone, cartilage, heart, kidney. How unique is MGP expression to LECs? What are known functions of MGP that could affect tumor cells? Is MGP expressed in other cells in the lymph nodes? In vitro assays showed that MGP inhibits tumor cell adhesion to LECs, however, further studies are needed to establish the importance of increased adhesion, if any, for metastasis. This cannot be assumed.

LEC expression is not unique to lymphatics but its constant upregulation in metastatic LNs of every single patient is noteworthy. As it is in numerous examples in the literature, studies in many areas have focused on limited aspects of identified molecules and that is reflected in

names of many molecules. Importantly, further studies have shown additional properties for those molecules. In case of MGP nothing is known about its role in lymphatics. Thus, its role in adhesion reported in this manuscript is completely new information.

Upregulation of MGP in sentinel nodes concerned lymphatics as despite expressed by the LN capsule, its expression in the capsule was not changed upon metastasis. Moreover, we conducted *ex vivo* adhesion assays also using distal, non-cancerous LNs. In these assays, anti-MGP antibody did not reduce binding being on line with its low expression in normal LNs. Importantly, we now demonstrate that recombinant MGP binds cancer cells. These results are now presented in Fig 8f and g. These new experiments show that MGP directly supports cancer cell binding and this takes place especially in metastatic LNs.

(For ethical reasons, patient studies are not currently possible. Thus, these *ex vivo* studies are closest, which are currently possible. These types of *ex vivo* assays have been highly accurate to reflect what happens *in vivo* concerning the leukocyte and cancer cell adhesion to vasculature. Alpha 4 integrins targeted by natalizumab and vedolizumab, both well selling drugs on market and inhibiting $\alpha 4/\beta 1$ and $\alpha 4/\beta 7$ binding to VCAM-1 and MAdCAM-1) are excellent examples, the functions of which were found by this assay method (Yednock et al. Nature 356, 63-66, 1992, Fig. 2 and Hu et al. PNAS 89: 8254-8258, 1992, Fig. 5). Examples on lymphatics are Clever-1 and macrophage mannose receptor, both of which mediate binding in *ex vivo* assays and are involved in trafficking within the lymphatics *in vivo* (Irjala et al. J Exp Med 194:1033-1041, 2001, Fig 2; Salmi et al, Circ Res 112:1577-1582, 2013, Fig. 3; Irjala et al. Cancer Res, 63: 4671-4676, 2003, Fig. 1 and Hollmén et al, BJC 123:501-509, 2020: anti-Clever-1 antibody is in clinical trials)).

4. In vitro studies examined the effects of tumor cell CM on LECs, however, it has not been proven that the observed changes are specific to tumor cells. Normal breast epithelial cells should be used as a control. LECs may change also in co-culture with normal epithelial cells.

To address this point, we have now conducted additional experiments using CM from tumor tissue and adjacent normal tissue explants from breast cancer patients. Although the observed changes were generally modest, the expression trend for our main gene of interest, MGP, was consistent with that seen in the original experiments. Some of the observed differences are likely attributable to the distinct secretion profiles of tissue explants compared to established cancer cell lines, supporting the interpretation that the originally observed changes are largely tumor cell-specific (Supplemental Figure 13).

5. Authors have indicated that LECs transition from immunologically active LEC subsets to those that are immunosuppressive. This is a strong statement that is based in transcriptomic data and needs to be backed up with direct evidence or the statements should be softened. Anti-inflammatory state of LECs may be induced because of a co-culture with (normal) epithelial cells, and may not be specific response to a tumor. Anti-inflammatory state is an overstatement in view of the data. presented, and would have to be better defined.

We have removed the part discussing this aspect.

6. On page 11, it is stated that "Despite existing beliefs that tumors induce proliferation of

LECs in murine models, we observed no difference in cell proliferation upon incubation with various media" (Supplemental Figure. 8). This statement needs to be corrected - not all tumor cells will induce lymphangiogenesis and it should not be expected that all tumor cells induce LEC proliferation in vitro. The text should be corrected to indicate that "the cell lines tested did not induce LEC proliferation in vitro under such and such conditions".

This has been modified as suggested.

7. Discussion includes many hypotheses that are not supported by the data. It should be revised to be less vague and closer to the data. There is a lot of hypotheses about the function of MGP. There are vague statements such as "This may be due to multiple factors in the CM causing different effects"; "We propose that cancer-promoting MGP expression in LECs has the potential to contribute to cancer behavior". Discussion should be streamlined.

We have removed these sentences. In addition, we have modified the section discussing CD200 and MGP, and shortened the discussion.

8. Discussion: There is a statement that in cell culture, LN LECs dedifferentiate and therefore do not accurately represent different in vivo clusters". That statement is too strong, while LECs may change their properties in the in vitro environment, they maintain lineage characteristics and do not dedifferentiate. This should be rephrased.

The referee is correct that certain lymphatic markers remain, while many phenotype markers are lost during the culture, especially in higher passages. The sentence has been modified (Lines 522-523).

Reviewer #3 (Remarks to the Author):

The authors demonstrate a significant shift in LEC composition and phenotype in response to extracellular components produced by breast cancer cells following colonization by metastatic cells - potentially by TGFbeta, VEGFA, and/or VEGFC which have well established roles in tumour microenvironment remodelling. Notably, this shift is associated with reduced inflammation / immunological functions which likely contributes to the immune-escape of metastatic cells. They present a large cohort of scRNAseq from patient-matched tumour-proximal metastatic lymph nodes and distal uncolonized lymph nodes, which revealed a metastatic-specific population of CD200+ HAY1+ LECs possibly driven by response to TGFb expression. In addition, they validate this finding by exposing in vitro LECs to media conditioned by breast cancer cell-lines using flow, qPCR and bulkRNAseq. They further show this shift (as characterized by upregulation of MGP) is inducible by TGFb, VEGFA and/or VEGFC and is associated with greater adherence of tumour cells to LECs supporting its role in colonization of lymph nodes by metastatic cancer cells. Overall I find the results convincing and comprehensive, and revealing a novel specific mechanism of tumour progression.

Major Concerns

The authors state that TGFBR2/TGFBR3 etc.. are expressed on CD200+LECs but the only plot (Figure 4f) provided to support this is of the prior weights used by NicheNet which

would not account for the expression in CD200+LECs, could the authors provide direct evidence that these receptors are expressed in the CD200+LECs?

In our single-cell dataset, all LEC subsets, including the CD200⁺ HEY1⁺ LECs, show significant expression of TGFBR2 and TGFBR3. We have included an expression plot highlighting receptors of TGF- β and VEGF across LN subsets. This new figure was added to Supplemental Figure 10.

The authors suggest that CD200+ LECs were near metastatic cancer cells, however in the provided figure (Figure 2g) they appear generally ubiquitously present both near and distant from the tumour cells. How did the authors quantify the proximity between different LEC populations to the metastatic cells? In addition, I had difficulty spotting the keratin+ tumour cells indicated by the white arrows.

We agree with the reviewer that CD200⁺ LECs, while predominantly present in metastatic lymph nodes, are not necessarily located in close proximity to tumor cells within the nodes. We have revised the corresponding description in the text to reflect this more accurately. Regarding Figure 2g, we also appreciate the comment about the visibility of keratin⁺ tumor cells. To improve clarity, we have changed the color of the tumor cells to make them more easily detectable. The updated figure is now included in the revised manuscript (Figure 2).

The authors state that VEGFA is highly expressed by macrophages, but in Figure 4e it is shown that cancer cells express VEGFA at a similar level.

We have now added that VEGFA is also expressed by cancer cells (Line 275).

For the trajectory analysis, what is the biological justification for ACK4+SCS ceiling LECs being the root of the trajectory?

During LN development, the subcapsular sinus (SCS) forms following the engulfment of the LN anlage by collecting lymphatic vessels (Bovay E, *J Exp Med*, 2018). This developmental sequence suggests that SCS ceiling LECs arise early in the formation of the LN lymphatic network. Consistently, our previous trajectory analysis using tSpace (Takeda A, *Immunity*, 2019) also identified ACKR4⁺ SCS ceiling LECs as the root of the trajectory, supporting their role as a precursor population in the LEC lineage hierarchy. This discussion was added in the lines 431-440.

Minor Concerns

In figure 1b what does AXLN1-4 mean?

Axillary lymph nodes. The abbreviation has now been spelled out in the Figure legend (Fig.1).

In figure 2e the authors postulate a developmental/differentiation relationship between the different LEC populations. What evidence is there for this? Monocle and most other pseudotime methods simply identify gradients in expression, those gradients do not necessarily represent developmental/differentiation relationships; they can equally represent spatial gradients arising from oxygen or other nutrient availability or tissue organization factors. CCA integration tends to generate gradient-like structures in its lower-dimensional

embedding, is there orthogonal evidence that LECs can transition between the identified cell-states?

We agree that pseudotime methods primarily identify transcriptional gradients, which may reflect not only developmental or differentiation trajectories but also spatial or microenvironmental influences. However, since the newly emerging subset (CD200⁺HEY1⁺LECs) is absent in distant LNs, we hypothesized that this population arises from transitions of existing LEC subsets found in distant LNs. Nonetheless, we acknowledge the limitation that pseudotime methods do not provide definitive evidence of differentiation. Future fate-mapping studies will be necessary to experimentally validate transitions between the identified LEC states. We have added this point to the Discussion, lines 438-440.

The authors state in their discussion that updating to Seurat v4 would offer significant improvement in integration over using CCA, but they state in their methods that they are using CCA integration in Seurat v4. They should clarify these contradicting statements.

In this study, we actually used reciprocal PCA (RPCA) within the Seurat v4 integration framework to identify anchors, which are pairs of biologically similar cells across datasets. These anchors were then used to integrate the datasets and correct for batch effects. In contrast, our previous study (Takeda et al, *Immunity* 2019) used Seurat v2.3, where integration was performed using diagonal canonical correlation analysis (CCA) (Butler et al., *Nat Biotechnol*, 2018). Unlike the CCA-based approach, which aligns datasets in low-dimensional space, the anchoring method used in Seurat v4 generates a fully integrated gene expression matrix, enabling more accurate downstream analyses such as clustering and trajectory inference. We corrected the descriptions in the Discussion, lines 423-426. We also included our code in the Code Availability section to enable others to reproduce our analysis.

The authors exclude 2 clusters due to concerns with data quality, while these are well justified, were these clusters associated with specific samples or metastatic or distant LNs? Tumours may trigger stress responses in tissues thus the poor quality of those clusters may not be due to technical artefact, but rather be a real biological effect of tumours on LECs.

One (cluster 3) shows high expression of mitochondrial genes such as MT-CO3, while another (cluster 4) is enriched for heat shock proteins such as HSPA6 (Figure 2). These clusters appear consistently regardless of LN metastasis (a), and patients (pt, b), and are most likely artifacts arising from technical factors. This description was added into the materials and methods sections (lines 615-617).

Figure 2. a, b, Clusters showing high expression of mitochondrial and heat shock protein genes, which are removed from further downstream analysis.

For the Monocle analysis does enabling the circular path option have a significant effect on the results? It appears that T2 and T6 could potentially be a circular (i.e. converging) trajectory. What is the justification for disallowing this option?

We also performed Monocle analysis with the circular path option enabled (close loop =TRUE; see figure below). While T2 and T6 remained, enabling this option resulted in T1 being connected to both T2 and T6 (Figure 3). However, in our main analysis, the close loop option was disabled, as we considered LEC differentiation in metastatic LNs to follow a one-directional progression of cell states. These statements were added to the materials and methods sections (lines 645-646).

Figure 3. Monocle analysis of LN LECs with the circular path option enabled

The authors should specify the version of NicheNet they are using, this package has been

revised multiple times since publication, thus the version number is necessary to ensure reproducibility of results.

We used nichenetr 2.1.0. We added this information to materials and methods.

Which methodology was used for pseudobulk DE for the scRNAseq data?

We separated the single-cell dataset into metastatic and non-metastatic lymph nodes and ran FindAllMarkers with default settings. This is described in the Materials and Methods section (lines 630-631).

Reviewer #4 (Remarks to the Author):

In this paper, Eichin et al. used single cell RNA sequence analysis to profile lymphatic endothelial cell (LEC) subsets in paired metastatic and non metastatic lymph nodes (LN) from 9 patients with luminal and HER2+ breast cancer. They found new LEC subsets in metastatic LNs and transcriptional changes. Matrix Gla protein (MGP) was one of the most upregulated genes in all LEC subsets across all patients. MGP expression in LECs was TGFb and VEGF dependent and regulated HLEC migration and cancer cell adhesion to LN lymphatics.

Overall, this is an interesting and very informative study that provides new knowledge on the transcriptional program modulated in human metastatic LN compared to non metastatic LN in breast cancer patients. The high-quality data are solid, convincing and well performed with appropriate controls and statistical analyses. However, the underlying mechanisms and the role of MGP in breast cancer metastasis to LN still remain poorly defined as it is in the present manuscript.

We have now added new experimental data concerning the function of MGP. First, we demonstrate direct binding between MGP and cancer cells using recombinant MGP (Fig. 8f). We have also now tested the role of MGP on non-metastatic lymph nodes and found that in line with its low expression, the anti-MGP antibody did not decrease tumor cell binding to lymphatic sinusoids in non-metastatic lymph nodes (Fig. 8g).

Major comments:

1) Figure 5: an HER2+ cell line (HCC1954, SKBR3) should be added to reflect the patients with luminal and HER2+ breast cancer. The rationale to use MDA-MB-231 cell line is not detailed in the manuscript.

Additional experiments investigating HCC1954 cells have been added to Figure 6f and e. We have now written that we included MDA-MB-231 cells to have a cell line representing triple negative breast cancer, thereby representing a more aggressive phenotype.

2) Figure 6: the rationale to test MGP as one of the most upregulated gene by the CM and in the patient LN is clear, however, it would add to test other genes pro-inflammatory (MGP, SOCS3, RAMP3) and anti-inflammatory (VCAM1, IL4) to support the hypothesis of a switch in LECs from a pro-inflammatory to anti-inflammatory state and the pathway analysis in Figure 5e.

We performed additional experiments to investigate the expression levels of RAMP3 and IL4I1 following various recombinant cytokine or antibody treatments. These results are shown in Supplemental Figures 16 and 17. The data indicate that the tested reagents differentially regulate the expression of these molecules, without exerting universal effects.

3) Figure 6b: what about the MCF-7 CM? it seems to be also reduced by the blocking antibodies.

Indeed, it does look like also MCF-7 CM is reduced by the blocking antibodies. However, due to the presence of an “outlier” this does not become statistically significant. If this datapoint is removed, significance would be reached. However, we decided not to exclude this datapoint and thereby statistical significance is not reached.

4) Figure 7e: the difference in HLECs migration between control siRNA and siMGP is very little and therefore, not very convincing to support a role of MGP in HLECs migration. Since figure 5 and 6 show the increased expression of MGP in response to breast cancer CM and VEGF/TGF β , it would be better to assess cell migration and invasion (boyden chamber assays), and adhesion assays in response to either breast cancer cell lines CM or with VEGF or TGF beta, with control siCtrl, siMGP, or anti-MGP antibodies.

We agree that the difference in migration is rather small, yet still statistically significant. To make it more transparent, we have extended the description of our statistical method used (Lines 1180-1181). In addition, we have performed Boyden chamber assays where we investigated either the migration/invasion of HLECs that had been silenced for MGP (siMGP) or with a control construct (siCtrl) through the trans-wells without and with Matrigel coating. We did not observe any difference in migration/invasion between silenced and control HLECs (Figure 4). Thus, based on the results obtained in the scratch wound assay, the impact of MGP seems to be in the lateral movement.

Figure 4. Boyden chamber assays for determining migration and invasion of HLECs

Regarding the cancer cell adhesion, we extended the *ex vivo* assays by including binding to non-metastatic LN lymphatics using anti-MGP and control antibodies (Fig. 8g). These assays most likely reflect best what takes place *in vivo*.

These additional experiments would help strengthen the proposed mechanistic model and the role of TGF β and VEGF-induced upregulation of MGP on breast cancer cell dissemination in patients, which could lead to combined therapeutic strategies for breast cancer patients.

Minor comments:

Figure 1b: not sure what is AXLN1, 2, 3, 4.

Axillary lymph nodes. The abbreviation has now been spelled out in the Figure legend (Fig.1).

Dear Reviewers,

Please find our revised manuscript attached. As this study addresses breast cancer, which occurs approximately 100 times more frequently in women than in men, it does not include male patients. Our detailed, point-by-point responses to the reviewers' remaining concerns are provided below and modifications are indicated in red in the manuscript. We hope that these revisions meet your expectations and that our manuscript will be considered suitable for publication.

Reviewer #1 (Remarks to the Author):

With the additional bioinformatics analysis, spatial transcriptomic data, and the new functional experiments, the authors addressed most of my questions and concerns, and the revised manuscript has been significantly improved.

I am convinced that the CD200+HEY1+ capillary-like LECs are enriched in metastatic LNs compared to distant non-metastatic LNs. Mechanistically, the authors propose the presence of MGP in this new population of LECs promotes cancer cell invasion in the LNs. However, in the Supplemental table 1 Excel file containing the up and down regulated genes in different clusters, I could not find the MGP gene in any of the clusters. And why all the avg_log2FC are negative? What are these upregulated genes? In figure 4 panel C, MGP is broadly expressed in many subclusters of LECs in both non-metastatic distant LNs and metastatic LNs as shown in the violin plot. That means MGP is not only restricted to the capillary-like CD200+HEY1+ LECs (cluster 4). For instance, in the violin plot, MGP is substantially increased in cluster 5 LECs in metastatic LNs as well. Any comments on the other MGP+ clusters of LECs? Furthermore, I am wondering, what other cell types in the metastatic LN also expressed MGP? Could the author check the MGP gene expression in the single-cell datasets?

We apologize for the confusion regarding the supplemental data. The Excel file contains two separate sheets: one listing upregulated genes (avg_log2FC > 0) and another listing downregulated genes (avg_log2FC < 0). As noted by the reviewer, MGP is indeed included among the top upregulated genes in several LEC clusters, including cluster 4 (please refer to the Excel file rather than to the PDF). This was also mentioned in line 75 of the manuscript. MGP is upregulated by multiple LEC subsets in metastatic LNs possibly due to VEGF and TGF signaling, and its upregulation—regardless of LEC subtype—may contribute to cancer metastasis. In our single-cell dataset, MGP expression is predominantly observed in LECs, BECs, and stromal cells within metastatic LNs (Supplemental Fig. 7a).

The new spatial transcriptomic data in figure 3 is very cool, it clearly showed the difference between LECs in distant LNs and metastatic LNs. Is it possible to include MGP in the dotplot in panel D? Also, in the metastatic LN from patient #3, it seems like many of the cells in "Cancer cells and LEC1" cluster are enriched in the SCS, any comments on this population? Was this LN at the very early stage of cancer metastasis that's why many of the cancer cells were enriched in the SCS? It seems like the spatial distribution of LECs in metLNs in Pt#3 and Pt#5 are very different.

We have now included MGP in the dot plot shown in Figure 3D, as requested. Regarding the metastatic LN from patient #3, we do not consider this sample to represent an early stage of metastasis, as numerous cancer cells are already present within the LN parenchyma (see below). Rather, this LN appears to be at a more advanced stage, lacking the typical architecture of SCS and medullary sinuses, with cancer cells disseminated throughout the node. Although LECs were detected near the capsule (now outlined by dotted lines in Fig.3) in patient #3, these structures do not correspond to the SCS, as shown by Fig.1 of this letter. The “Cancer cells and LEC1” cluster closely resembles capillary-like LEC 1 or 2, but these cells also express multiple cancer cell markers, such as CD24, suggesting interactions between LECs and cancer cells in this region (please note that Visium HD is not single-cell resolution). The spatial differences in LEC distribution between patients #3 and #5 are described in the Results section (line 205-207).

Figure 1. Visium HD analysis of cellular subsets in metastatic LN from patient #3. Arrows indicate lymphatic vessels beneath the capsule. The metastatic LN from patient #3 displays a disrupted architecture and lacks the typical SCS structure observed in non-metastatic LNs.

Some minor comments:

Figure 1d and Figure 2a are a little bit redundant, consider move one of them to the supplemental figures. In figure 2a the left panel, there are very few cells from cluster #4, but the label of cluster #4 is missing.

We moved Figure 2a, b to the supplemental Figure 4. We also added the label of cluster 4 into the new supplemental Figure 4a.

In Figure 6c the heatmap, are these little triangles on the right side and the bottom have some specific meaning? Or maybe some labels are missing?

The triangles in Figure 6c are indicating the outcome of the unsupervised clustering, thereby showing which genes and which cell lines behaved in a similar way. They are merely a visual aid and do not have any labels. We have added this clarification to the figure legend and also cleaned up the figure.

Reviewer #2 (Remarks to the Author; the numbering of the comments is the original one):

Eichin et al. addressed most of the concerns raised. Several issues remain to be addressed, mainly related to data presentation and interpretation.

1. RE-REVIEW: The inclusion of Visium HD data further strengthens the study. In the Results section (lines 186-214), please indicate specifically how the Visium HD data align and do not align with the key scRNA-seq data.

As described in the Results section, the Visium HD data clearly recapitulate the key findings from the scRNA-seq dataset—specifically, the presence of SCS floor and medullary sinus LECs in non-metastatic lymph nodes, and the loss of these populations along with the emergence of capillary-like LEC subtypes in metastatic LNs. However, not all LEC subsets and subset-specific gene signatures identified by scRNA-seq (such as HEY1) were detectable in the Visium HD dataset. These differences may be attributed to the lower transcript capture efficiency and sequencing depth of Visium HD, as well as the analysis being limited to a single 10 μ m tissue section, rather than to single-cell resolution. These statements were added to the Result section (lines 213-216).

It should be mentioned in the Results that metastases in the LNs were large and that this markedly altered architecture of the entire lymph node. The assumption of the reader is that the lymph node architecture is preserved in presence of metastases, which it can be when metastases are smaller. Clarifying that metastases are large and that LN architecture is markedly changed is important for data interpretation.

Thank you for this important point. We have clarified in the Results section that the metastatic lymph nodes in our study contain large tumor burdens and exhibit markedly altered architecture (lines 194-195).

2. RE-REVIEW: Please show evidence that LECs used in this experiment were CD200+ (if shown already, indicate in the Results where has it been shown). In the corresponding Methods section, clarify which LECs specifically were used in the experiments with T cells.

Commercially obtained or isolated by your lab, and from what source. Cell lines or primary cells. Several sources are indicated in the methods at the beginning.

We have added a new figure showing CD200 expression on the HLECs used (Supplemental Figure 5a). In addition, we have included the information about which HLECs were used in this experiment in the Methods section (cells obtained from CellBiologics (#H6092) and cells we had extracted ourselves), starting line 819.

4. RE-REVIEW: Studies shown in Fig. 6 were done using breast cancer cell lines. Primary breast epithelial cells or non-tumorigenic breast epithelial cell line would have been the most relevant control for this experiment. New data shown in Suppl. Fig. 13 using CM from tissue samples show very different results from what is shown in Fig. 6 using cell lines. Data shown in Fig. 6f has not been replicated with the new data in Suppl. Fig. 13. However, this is expected, since the tissues are complex and are expected to have a very different secretome than cultured cell lines, which authors acknowledge.

However, the claim that "the originally observed changes are largely tumor cell-specific" is misleading, because it suggests that non-tumorigenic cells do not induce such changes, and that has not been demonstrated. If the authors want to determine what gene expression changes in LECs are specifically induced by tumor cells, then non-tumorigenic breast epithelial cells should be used as a control. Alternatively, if the data remains as it is now, interpretation needs to be modified - it cannot be stated that the changes are tumor-specific. Factors produced by tumor cell lines induced these changes in LECs, but this does not mean that they are tumor-specific, or that the same will be seen in vivo, as evidenced by the recent experiment.

The challenge with using so-called normal epithelial cell lines is that these cells have been genetically altered to grow indefinitely, which means they no longer fully represent normal tissue. We therefore concluded that explant tumors and adjacent normal tissues provide a much closer reflection of real-life biological condition. For this reason, we chose to focus on explant tumor models.

Thank you for pointing your concern out. However, we did not include the statement that "the originally observed changes are largely tumor cell specific" in the manuscript itself; it was only mentioned in our earlier response letter. To improve clarity, we have now revised the manuscript to include the following explanation: "Moreover, CM of cancer tissue explants resulted in different gene expression profiles than the CM of adjacent normal breast tissue, including a trend toward upregulation of MGP (Supplemental Fig. 13), suggesting that soluble factors in the tumor microenvironment influence LEC phenotype" (main text, lines 333–335).

Suppl. Fig. 13, n=6 means replicates from the same sample for PCR (technical) or CM from 6 different LN samples?

CMs are from 3 different explant samples, all tested with two different HLECs. Now this is indicated in the Figure legend.

In Fig. 6, please indicate in the Figure legend the time - how long were LECs incubated with tumor CM (48hr?) and for how long was media conditioned (24hr?). It does not matter on which days, but the duration.

Done

7. RE-REVIEW: Discussion, Lines 410-411: "MGP silencing/inhibition increased LECs migration and reduced cancer cell adhesion" change to be more clear. Suggestion: "MGP silencing or inhibition increased LECs migration and reduced cancer cell adhesion, respectively."

Done

Discussion, Lines 441-443: "Expansion of this subset in metastatic LNs provides the cancer cells with an improved opportunity to. leave the LNs and migrate systemically" There is no evidence that this subset supports egress of tumor cells, this statement should be softened. Suggestion: "...may provide with an opportunity"

Done

Discussion, Lines 447-450: "Hence, our study indicates that tumor metastasis within LNs instigates a transition from immunologically active LEC subsets....to those that are immunosuppressive and conducive to tumor metastasis". This is an oversimplification, please reformulate. "Immunologically active". Previous studies have shown that LECs in lymph nodes promote peripheral tolerance; studies by Engelhard for example have shown complexities of lymph node LEC contribution to immunity.

We have modified the sentence: ...a transition from existing LEC subsets towards those carrying markers for immunosuppressive phenotypes.

Reviewer 4

Minor corrections:

Line 294: HER2+

Line 374: Fig. 8a

Performed